

# Surface processes and drivers of the snow water stable isotopic composition at Dome C, East Antarctica – a multi-datasets and modelling analysis

Inès Ollivier[1,2], Hans Christian Steen-Larsen[1], Barbara Stenni[3], Laurent Arnaud[4], Mathieu Casado[2],

Alexandre Cauquoin[5], Giuliano Dreossi[3], Christophe Genthon[6], Bénédicte Minster[2], Ghislain Picard[4],

Martin Werner[7] and Amaëlle Landais[2]

[1]Geophysical Institute, University of Bergen, and Bjerknes Centre for Climate Research, Bergen, Norway
[2]Laboratoire des Sciences du Climat et de l'Environnement, LSCE/IPSL, CEA-CNRS-UVSQ, Université Paris-Saclay, Gif sur Yvette, France
[3]Department of Environmental Sciences, Informatics and Statistics, Ca' Foscari University of Venice, Mestre Venice, Italy
[4]Université Grenoble Alpes, CNRS, INRAE, IRD, Grenoble INP, IGE, Grenoble, France
[5]Institute of Industrial Science, The University of Tokyo, Kashiwa, Japan
[6]Laboratoire de Météorologie Dynamique, LMD/IPSL, Sorbonne Université-CNRS, Paris, France
[7]Alfred Wegner Institute (AWI), Helmholtz Centre for Polar and Marine Research, Bremerhaven, Germany

*Correspondence to*: Inès Ollivier (ines.ollivier@uib.no)

**Abstract.** Water stable isotope records in polar ice cores have been largely used to reconstruct past local temperatures and other climatic information such as evaporative source region conditions of the precipitation reaching the ice core sites. However, recent studies have identified post-depositional processes taking place at the ice sheet's surface modifying the original precipitation signal and challenging the traditional interpretation of ice core isotopic records. In this study, we use a combination of existing and new datasets of the precipitation, snow surface and subsurface isotopic compositions ($\delta^{18}O$ and d-excess), meteorological parameters, ERA5 reanalyses, outputs from the isotope-enabled climate model ECHAM6-wiso, and a simple modelling approach to investigate the transfer function of water stable isotopes from precipitation to the snow surface and subsurface at Dome C, in East Antarctica. We first show that water vapor fluxes at the surface of the ice sheet result in a net annual sublimation of snow, from 3.1 to 3.7 mm water equivalent per year between 2018 and 2020, corresponding to 12 to 15% of the annual surface mass balance. We find that the precipitation isotopic signal cannot fully explain the mean, nor the variability of the isotopic composition observed in the snow, from annual to intra-monthly timescales. We observe that the mean effect of post-depositional processes over the study period enriches the snow surface in $\delta^{18}O$ by 3.3‰ to 6.6‰ and lowers the snow surface d-excess by 3.5‰ to 7.6‰ compared to the incoming precipitation isotopic signal. We also show that the mean isotopic composition of the subsurface snow is not statistically different from that of the surface snow, indicating the preservation of the mean isotopic composition of the surface snow in the top centimetres of the snowpack. This study confirms previous findings about the complex interpretation of the water stable



isotopic signal in the snow and provides the first quantitative estimation of the impact of post-depositional processes on the snow isotopic composition at Dome C, a crucial step for the accurate interpretation of isotopic records from ice cores.

## 1 Introduction

Polar ice cores have been widely used in paleoclimate studies to reconstruct past atmospheric conditions, up to 800 000 years back in time (EPICA community members 2004). Within the ice matrix of the core, $\delta^{18}O$ and $\delta D$ (Craig, 1961) measurements have been commonly used as a proxy for past atmospheric temperatures based on the observed relationships between the local atmospheric temperature and both the isotopic composition of precipitation samples (Dansgaard, 1964) and the snow across spatial transects in Antarctica (Lorius et al., 1969, Masson-Delmotte et al., 2008).

The second order parameter deuterium excess (d-excess), defined as the deviation from the existing linear relationship between $\delta^{18}O$ and $\delta D$ (d-excess = $\delta D - 8 \times \delta^{18}O$, Dansgaard, 1964), is driven by physical processes involving non-equilibrium, or kinetic fractionation of the different isotopes. The d-excess measured in ice cores has been interpreted as a proxy for moisture origin (Masson-Delmotte et al., 2005) and conditions at the moisture source region, such as sea-surface temperatures and relative humidity above the ocean's surface (Merlivat and Jouzel, 1979; Jouzel et al., 1982; Vimeux et al.,

1999; Stenni et al., 2001; Uemura et al., 2008, 2012; Landais et al., 2021; Steen-Larsen et al., 2014a). Further kinetic processes along the distillation path of an air mass have been identified to contribute to the d-excess signal in precipitation, such as condensation in supersaturated conditions (Jouzel and Merlivat, 1984) or mixing of air masses from different origins (Risi et al., 2013).

The reconstruction of the climatic parameters from the water isotopic records in polar ice cores relies on the assumption that the isotopic composition of precipitation is preserved from snowfall to burial and transformation into ice. However, this has been challenged by recent field studies highlighting the significant role of post-depositional processes at the surface of both the Greenland and Antarctic ice sheets modifying the isotopic composition of precipitation after snowfall (Touzeau et al., 2016; Münch et al., 2017; Casado et al., 2018, 2021b; Hughes et al., 2021; Wahl et al., 2021, 2022; Zuhr et al., 2023). The

post-depositional processes commonly proposed to affect the water isotopes at the ice sheet's surface include (i) water vapor exchanges between the snow and the lower atmosphere through sublimation and condensation cycles, (ii) wind redistribution and (iii) diffusion of water vapor within the snowpack.

On the Greenland Ice Sheet, Steen-Larsen et al. (2014b) provided the first evidence of a co-variation of the snow surface and the low atmosphere water vapor isotopic compositions during precipitation-free periods in the summertime, suggesting

seasonal vapor exchanges between the snow and the atmosphere. Wahl et al. (2021) later measured a depleted sublimation humidity flux compared to the snow surface, showing that fractionation of water isotopes was taking place during sublimation. Including fractionation during sublimation in a simple model also improved the prediction of the day-to-day variability in the snow isotopic composition during summertime (Wahl et al., 2022). Additional laboratory and modelling



studies showed that sublimation leads to an enrichment in $\delta^{18}$O together with a lowering of d-excess in the surface snow and in the firn (Hughes et al., 2021; Dietrich et al., 2023).

In addition, diffusion of water vapor within the snowpack is driven by temperature and isotopic gradients and affects the isotopic composition of the snow and firn continuously (Johnsen et al., 2000; Gkinis et al., 2014). Field studies on both the Greenland and Antarctic Ice Sheets identified snow metamorphism associated with water vapor diffusion within the top layers of the snow to affect the snow isotopic composition ($\delta^{18}$O and d-excess) after snowfall (Casado et al., 2021b; Harris Stuart et al., 2023).

Lastly, the wind blowing at the surface of the ice sheet leads to a heterogeneous accumulation by redistributing the snow on the surface (Libois et al., 2014; Picard et al., 2019; Zuhr et al., 2021) which impacts the build-up of the isotopic signal in the snow (Zuhr et al., 2023). Wind is also hypothesized to impact the snowpack isotopic composition through forced pumping and ventilation of the snowpack (Town et al., 2008).

At Dome C, on the East Antarctic Plateau, previous studies have focused on qualitative description of the impact of post-depositional processes on the snow surface at Dome C (Casado et al., 2018), monitoring the atmospheric water vapor, surface snow and precipitation isotopic compositions (Casado et al., 2016; Touzeau et al., 2016; Stenni et al., 2016; Dreossi et al., 2023) or exploring the isotopic signature of snow metamorphism (Casado et al., 2021b). However, a comprehensive understanding of the formation of the isotopic signal in the snow, is still missing.

In this study we address the transfer function of water stable isotopes from precipitation to the snow surface and subsurface at Dome C, from intra-monthly to multi-annual timescales. We use a combination of existing and new datasets of the isotopic composition of precipitation, surface snow and subsurface snow over five consecutive years (2017-2021), ERA5 reanalysis products, outputs from the isotope-enabled climate model ECHAM6-wiso and a simple modelling approach to investigate the contribution of precipitation to the variability observed in the snow surface and subsurface isotopic composition. In addition, we use the meteorological parameters measured continuously on-site to estimate the magnitude of sublimation and condensation fluxes between the surface and the lower atmosphere over three consecutive years (2018-2020) and qualitatively evaluate their impact on the snow isotopic composition at Dome C.

## 2 Data and methods

### 2.1 Geographical settings

Dome C is located on the East Antarctic Plateau (75.1°S, 123.3°E), at 3233 m a.s.l. and 1000 km from the coast and is the site where the permanent research station Concordia is installed (see location on Fig. 1a). The site is characterized by a mean annual temperature of about -52°C (Genthon et al., 2021a) and a low accumulation rate of about 2.5 cm w.e. year$^{-1}$ (Genthon et al., 2015). Due to the very small local slope and the location on a dome, the site is not subjected to strong katabatic winds, the mean annual wind speed close to the surface is about 4 m s$^{-1}$ (Genthon et al., 2021a).



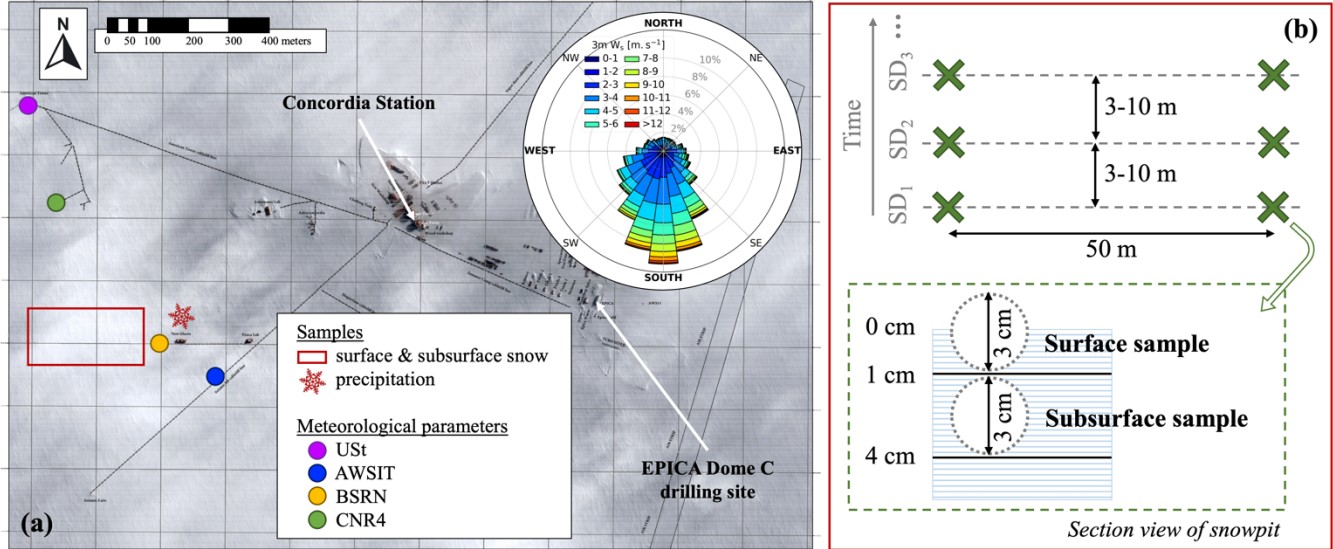

**Figure 1.** (a) Aerial view of Concordia station. The coloured circles indicate the location of the meteorological measurements used in this study (Sect. 2.4) and in red (rectangle and snowflake) the location of the samples presented in this study (Sect. 2.2 and 2.3). The wind rose for the five-year period 2017-2021 of the wind at 3 m is shown in the upper right corner. Background image from CNES (Pléiade satellite image of Concordia Station, Antarctica, CNES 2016, Distribution Airbus Defence and Space). (b) Snow sampling scheme taking place in the red rectangle in panel (a) (described in Sect. 2.2). SD stands for sampling day.

## 2.2 Snow surface and subsurface sampling

The regular sampling of the top few centimetres of the snow started in November 2013 and the sampling of a subsurface layer was added in 2017. Since then, the sampling strategy remained the same. In this study we focus on the 5-year period 2017-2021, where both surface and subsurface samples are available for analysis (see also Table 1).

The sampling takes place in the clean area about 800 m upwind of the main buildings (see location on Fig. 1a), twice a week and all-year round. The samples are taken at two different locations 50 m apart along a straight line, 3 to 10 m next to the line of the previous sampling day. At each location along the line, a small vertical snowpit is dug (about 20 cm deep). The snow surface and subsurface are collected with two 50 mL Corning tubes, placed horizontally from the snowpit wall, at the surface and just below. The two snow samples correspond to depths of 0 to 1 cm deep for the surface sample and from 1 to 4 cm for the subsurface sample, although the exact sampling depths was not recorded and may have slightly varied with the change of operator throughout the years or within one year due for instance to hard snow. The sampling scheme is illustrated in Fig. 1b.

Once the snow samples are collected, the tubes are sealed to prevent air exchange with the surrounding atmosphere and stored at temperatures well below freezing. The samples are shipped back once a year to LSCE (CNRS Paris-Saclay) to measure their isotopic composition with a laser spectrometer PICARRO L2130-i in liquid mode. We report the snow isotopic composition with delta-notation in ‰ (Craig, 1961) with respect to the Vienna Standard Mean Ocean Water



(VSMOW) (Gonfiantini, 1978). The associated uncertainty (two standard deviation of sample replicates) on these measurements is ± 0.2‰ for $\delta^{18}$O and ± 0.7‰ for $\delta$D.


The 5-year compiled dataset of the snow surface and subsurface isotopic composition used in this study is in (dataset in review with PANGAEA). Part of this dataset was published in Casado et al. (2021a), with an overlap with this study from January 2017 to April 2018 for the surface snow samples.

## 2.3 Precipitation sampling

Since 2008, precipitation samples have been collected in the vicinity of Concordia station, as part of different projects operated by the Italian Antarctic Research Programme (PNRA). Part of this long time series have been published in Stenni et al. (2016) (2008-2010) and Dreossi et al. (2023) (up to 2017). Here we extend the record of the precipitation isotopic composition to 2021 and use the timeseries from 2017 to 2021 for our analysis (see also Table 1). The new dataset (2018-2021) is available in PANGAEA (dataset in review with PANGAEA).

Precipitation samples are collected every day on a wooden platform (bench) 1 m above the ground, covered by a PTFE surface and shielded by an 8 cm rail. The bench is situated about 800 m upwind of the main buildings (see location in Fig. 1a). The samples are collected at 10 AM local time (UTC+8), although it has varied throughout the years depending on the operator. All the snow laying on the bench is collected, whether it is precipitation (including diamond dust), blown snow or air hoar from atmospheric condensation. It cannot be ruled out that some of the samples might have undergone sublimation

especially during the summertime, because of exposure to 24-hour solar radiation before sample collection (Stenni et al., 2016; Dreossi et al., 2023). Each precipitation sample collected is weighted, and we use these weights as approximate estimates of the precipitation amounts (details in Sect. 2.5).

After collection, the plastic bag containing the sample is sealed and stored at temperatures well below freezing before annual shipment to the Ca'Foscari University of Venice, Italy, where the isotopic composition of the samples is measured with a

PICARRO laser spectrometers (L2130-i and L2140-i). We report the snow isotopic composition with delta-notation in ‰ (Craig, 1961) with respect to the Vienna Standard Mean Ocean Water (VSMOW) (Gonfiantini, 1978). The associated uncertainty (one standard deviation of quality standard replicates) on these measurements is ± 0.08‰ for $\delta^{18}$O and ± 0.68‰ for $\delta$D.

## 2.4 Meteorological parameters

### 2.4.1 Atmospheric monitoring

Atmospheric parameters are measured continuously at Dome C by different weather stations and instruments installed nearby Concordia station. For the 2017-2021 period of interest of this study, we used meteorological observations both from



an Automatic Weather Station operated by the PNRA (referred as AWSIT hereinafter) and from a 42 m meteorological tower (referred as USt hereinafter). A summary of the meteorological parameters used in this study is available in Table 1.

The AWSIT is located about 800 m upwind of Concordia station (see location in Fig. 1a) and has been operating since 2005. The data are available at 1 hour time step (1-hour averages) in Grigioni et al. (2022). In this study, we use the atmospheric pressure measured at 1 m above the surface with a Vaisala PTB100 and a 3-month period of the atmospheric temperature measured at 1.5 m by a Vaisala HMP45D. To match the same time step of the observations from the USt described below, we linearly interpolated the data to 30 min.

The USt is part of the CALVA project and located about 1 km of Concordia station (see location in Fig. 1a). Meteorological instruments are installed at six different levels in the atmosphere measuring continuously for more than ten years (Genthon et al., 2021a). Due to snow accumulation, the sensors installed on the lowest level of the tower, at 3 m above the surface, were 40 cm closer to the surface in 2021 than in 2017. This height change was not considered here, and the lowest level of the tower is referred to as the 3 m level. All atmospheric parameters are sampled at 30 s intervals, however in this study we use 160 the 30 min averages.

The atmospheric temperature is measured by a PT100 in a Vaisala HMP155 combined sensor (thermohygrometer) placed in a mechanically aspirated shield (Young 43502) and the wind speed and direction are measured by Young 05103 aerovanes. The quality-controlled (QC) data for temperature and wind speed from 2010 to 2019 is available in Genthon et al. (2021b, c). Here we use this dataset from 2017 to 2019 and extend the record up to 2021 using the data from the same instruments 165 available on the CALVA project website (see Table 1). Since the quality-control of this additional period (2019-2021, referred as non-QC) is not guaranteed, we compared the QC and non-QC datasets during an overlapping period (not shown). We found no significant difference and therefore use the temperature and wind records from the USt between 2017 and 2021 in our analysis. Note that to fill a 3-month period of missing temperature data from the USt in 2021 (August to October), we use the temperature measured by the AWSIT (described above). There was no significant difference between the two 170 temperature records during overlapping periods (not shown). The mean temperature and wind speed over the period 2017-2021 are summarized in Table 1. The dominant wind direction over the same period is shown in the windrose in Fig. 1a.

Atmospheric water vapor content (or humidity) is also measured continuously by sensors installed on the meteorological tower. At Dome C, the surface atmosphere is very cold and frequently above saturation (Genthon et al., 2017), and in these conditions the standard humidity sensors fail to accurately measure the true atmospheric humidity content. To cope with this 175 issue, a modified HMP155 was designed and installed on the USt at 3 m above the ground, and proved its utility to measure atmospheric moisture accurately, capture supersaturation conditions and expand the temperature operating range of the humidity sensor (Genthon et al., 2017, 2022). The sensor reports atmospheric humidity with respect to liquid ($RH_{wrtl}$) water even at temperature below 0°C, we therefore convert $RH_{wrtl}$ to the relative humidity with respect to ice ($RH_{wrti}$), as in Genthon et al. (2017) and Vignon et al. (2022) (see also Supplement S1). The relative humidity at 3 m in the atmosphere and 180 at 30 min resolution over the 2018-2021 period is available in Genthon et al. (2021d) and an analysis of the dataset is available in Genthon et al. (2022). Vignon et al. (2022) further provides estimations of the uncertainties for $RH_{wrti}$ associated



with the temperature and humidity measurements. In this study, we use this 3-year atmospheric humidity record to estimate water vapor fluxes at the snow surface. The method is described in the following section. The mean $RH_{wrti}$ between 2018 and 2020 is indicated in Table 1.


**Table 1.** Summary of samples and meteorological parameters used in this study and the reference to the data. The average values for temperature and wind speed are calculated over the 2017-2021 period; the relative humidity with respect to ice, the atmospheric pressure and the surface temperature are calculated over 2018-2020. The CALVA project website is accessible at: https://web.lmd.jussieu.fr/~cgenthon/SiteCALVA/CalvaData.html. The CNR4 (*) data will be published upon acceptance of the article.

| Project | Type/depth or height | Sampling rate/measurement time step | Average | Reference to dataset |
|---|---|---|---|---|
| NIVO | Snow samples: surface (0-1 cm) and subsurface (1-4 cm) | 2x / week | – | Casado et al., (2021a) & this study |
| PRE-REC/ WHETSTONE | Precipitation samples | daily | – | Dreossi (2023) & this study |
| CALVA (USt) | 3 m temperature | 30 min | -52.1°C | Genthon et al. (2021b) & CALVA website (see Table caption) |
| CALVA (USt) | 3 m wind speed | 30 min | 3.9 m s$^{-1}$ | Genthon et al. (2021c) & CALVA website (see Table caption) |
| CALVA (USt) | 3 m $RH_{wrti}$ | 30 min | 104.5% | Genthon et al. (2021d) |
| AMCO (AWSIT) | Pressure at 1 m | 1 h | 642.6 hPa | Grigioni et al. (2022) |
| NIVO (CNR4) | Surface temperature | 10 min | -55.4°C | * see Table caption |

**2.4.1 Estimation of water vapor fluxes**

During the period of interest of this study, no direct Eddy-Covariance (EC) fluxes measurements were available at Dome C. We instead make use of the standard atmospheric parameters measured on site (described in the previous section) to apply the bulk method as described in Genthon et al. (2017) and estimate water vapor fluxes between the surface and the 3 m atmospheric level. We report the 30 min averages vapor fluxes during the 3-year period 2018-2020 in mm water equivalent
per timestep.

The bulk method is based on the Monin-Obukhov (MO) Similarity Theory (Monin and Obukhov, 1954) and relies on several assumptions, which may not hold over the Antarctic Plateau (Vignon et al., 2016). Nevertheless, this method is still commonly used as the parametrization of surface turbulent fluxes in global and regional climate models (e.g. MAR model, Gallée and Schayes 1994) and have been compared against Eddy-Covariance measurements both at Dome C on the East
Antarctic Ice Sheet (sensible heat fluxes, Vignon et al., 2016) and at EastGRIP on the Greenland Ice Sheet (water vapor fluxes, Dietrich et al., 2024). It requires the following parametrizations: (1) the choice of roughness length for momentum ($z_0$), (2) the choice of functions representing the atmospheric stability and (3) the calculation of the roughness lengths for water vapor ($z_{0q}$) and heat ($z_{0t}$). In their sensitivity study on the parametrization for sensible heat flux estimations at Dome C, Vignon et al. (2016) recommend the use of the stability functions from Holtslag and De Bruin (1988, referred hereinafter as



H88) for stable cases and the functions from Högström (1996) for unstable cases. They also recommend the use of a constant $z_0$ of 0.56 mm, which corresponds to the average value observed with an EC system over one year at Dome C (Vignon et al., 2016). For the parametrization of $z_{0q}$ and $z_{0t}$, we use the same approach as in Genthon et al. (2017) and King et al. (2001) where $z_0 = z_{0q} = z_{0t}$. We used this parametrization (H88, $z_0 = z_{0q} = z_{0t} = 0.56 \times 10^{-3}$ m) as the reference parametrization. In addition, and similarly as in Vignon et al. (2016) and Genthon et al. (2017), we computed the water vapor fluxes using three

other stability functions for stable conditions and a range of $z_0$ to estimate the sensitivity of the final flux calculations on the parametrization (fluxes computed 16 times, see Table 2 for stability functions and range of $z_0$). The range of $z_0$ tested corresponds to the observed range over one year at Dome C (Vignon et al., 2016).

**Table 2.** Set of roughness lengths for momentum $z_0$ and stability functions for stable conditions used to compute water vapor fluxes with
the bulk method. The reference parametrization is highlighted in bold.

| Roughness length for momentum $z_0$ | $0.01 \times 10^{-3}$ m |
|---|---|
| | **$0.56 \times 10^{-3}$ m** |
| | $1 \times 10^{-3}$ m |
| | $6.3 \times 10^{-3}$ m |
| Stability function for stable conditions | **Holtslag and De Bruin (1988) – H88** |
| | Lettau (1979) |
| | Grachev et al. (2007) |
| | King and Anderson (1994) |

To compute the water vapor fluxes, the bulk method requires temperature, wind speed, specific humidity, and pressure at the chosen atmospheric level (3 m here), as well as the snow surface temperature and the specific humidity at the surface.

For the atmospheric level, we use the temperature, wind and humidity sensors installed at 3 m above the surface on the USt together with the atmospheric pressure measured by the AWSIT (measurements described in Sect. 2.4.1). The formulas from Murphy and Koop (2005) are used to convert $RH_{wrti}$ into specific humidity. To guarantee that the stationary conditions required to apply the bulk method are met, we removed all 30 min temperature and wind speed data for which the differences in temperature and wind speed with the previous half-hour were above 2°C and 1.1 m s$^{-1}$, respectively (Vignon et

al., 2016). This represents 4% of the whole dataset.

For the surface level, the snow surface temperature is computed from upward and downward longwave radiative fluxes with the same method as in Vignon et al. (2016) (their Eq. 1), using the same snow emissivity of 0.99 (value given by Brun et al., 2011, used in Vignon et al., 2016 and Genthon et al., 2017). We use the longwave radiative measurements from a CNR4 radiometer installed approximately 500 m away from the USt (see location in Fig. 1a). We average the data over 30 min to

match the time resolution of the atmospheric measurements (originally 10-min resolution). Note that to fill a 3-month period of missing data in the CNR4 record at the end 2020, we use the data provided by the BSRN network (Lupi et al., 2021). The BSRN data was corrected on the CNR4 data during overlapping periods beforehand, due to a shift identified in the upward longwave flux measured by the BSRN pyrgeometer from December 2019 onwards. The mean surface temperature over the



2018-2021 time-period is reported in Table 1. The specific humidity at the snow surface is converted from the surface
temperature using the formulas from Murphy and Koop (2005) and assuming saturation.

In total, due to gaps in the different input datasets and the removal of non-stationary data, the missing data in the bulk
estimations represents 9% of the whole dataset.

## 2.5 Snow Isotopic Signal Generator (SISG)

To evaluate the contribution of precipitation to the isotopic variability observed in the snow surface and subsurface samples
collected at Dome C (described in Sect. 2.2), we use a simple modelling approach to create synthetic snow layers solely
based on the incoming precipitation. This approach was used in Casado et al. (2018, 2021b) for a similar purpose, but
focused on the top layer of the snowpack. Here we re-implemented the same simple model and added the subsurface snow.

The model (referred to as the Snow Isotopic Signal Generator – SISG) simulates snow layers by stacking precipitation events
until the thickness of the stacked precipitation reaches the depths of the snow surface and subsurface layers given as input of
the model. The isotopic composition of each snow layer is then calculated as the weighted average (by precipitation
amounts) isotopic composition of all precipitation events necessary to build the snow layers. We choose the input snow
layers depths to be 0 to 1 cm for the surface layer and 1 to 4 cm for the subsurface layer to match the snow samples collected
at Dome C. We run the model at daily resolution over the 5-year period 2017-2021 and retrieve the model results for the
same days as the observations. Table 5 summarizes the five model experiments performed with different inputs for the
precipitation isotopic composition and precipitation amounts (described in Sect. 2.5.1 and 2.5.2).

**Table 3.** Experiments performed with the SISG model, with respective inputs for the daily precipitation amounts (described in Sect. 2.5.1)
and the daily isotopic composition of precipitation (described in Sect. 2.5.2). Note that the precipitation amounts from the observations,
ERA5 and ECHAM6-wiso are scaled to match the mean annual accumulation observed at Dome C.


| Experiment | Daily precipitation amount | Daily precipitation isotopic composition |
|---|---|---|
| "Iso from T & cst accu" | Constant | Assuming constant iso vs temperature relationship |
| "Iso from T & ERA5 accu" | ERA5 | Assuming constant iso vs temperature relationship |
| "Iso from wg. mm & obs accu" | Observations | Precipitation-weighted mean annual isotope cycle |
| "Iso from ar. mm & obs accu" | Observations | Arithmetic mean annual isotope cycle |
| "Iso from ECHAM6 & ECHAM6 accu" | ECHAM6-wiso | Modelled by ECHAM6-wiso |

## 2.5.1 Daily precipitation amounts

The first SISG experiment uses a timeseries with a constant daily precipitation amount, calculated by dividing the mean
annual accumulation at Dome C by 365 days. We use a mean annual accumulation of 2.5 cm w.e. year$^{-1}$ estimated from stake
measurements (Genthon et al., 2015). This corresponds to 8 cm year$^{-1}$ using a snow density of 320 kg m$^{-3}$, a typical value for
surface snow at Dome C (Picard et al., 2014; Genthon et al., 2015; Leduc-Leballeur et al., 2017). Hereinafter, we use this





same snow density to convert precipitation amounts from snow water equivalent (SWE in mm w.e.) to snow depths (mm of snow) and inversely.

The second experiment uses the precipitation amount timeseries from ERA5 reanalysis (Hersbach et al., 2020, dataset available in Muñoz Sabater, 2019). We use the 24h-average of the hourly snowfall rate data for the grid point nearest Concordia station as an input to the model.

The third and fourth experiments use the observed precipitation amounts. As in Kopec et al. (2019), we assume that the weight of the daily precipitation samples collected on site for isotopic analysis (Sect. 2.3) is proportional to the amount of precipitation that has fallen over the day.

The fifth experiment uses the precipitation amounts given by the isotope-enabled global circulation model (GCM) ECHAM6-wiso (described in Cauquoin et al., 2019). The simulation was performed at a spatial resolution of 0.9° and nudged to ERA5 reanalyses (Cauquoin and Werner, 2021). The daily precipitation amounts were extracted for the grid point closest to Dome C. The ECHAM6-wiso outputs from 1990 to 2020 are available in Cauquoin and Werner (2023), and the simulation was extended to 2021 by A. Cauquoin.

Because the cumulative sum of the precipitation amounts according to the observations, ERA5 and ECHAM6-wiso was too low (18, 18, and 32 cm of snow after five years, respectively) compared to the mean accumulation over five years at Dome C (40 cm of snow), we scaled up all daily precipitation amounts in the three timeseries to match the mean annual accumulation (8 cm year$^{-1}$), acknowledging that this value is still underestimated because of net annual sublimation (Sect. 3.1). These scaled timeseries were used as inputs for the SISG model. The comparison of the three timeseries is shown in Fig. 5 (Sect. 3.3.1).

**2.5.2 Daily precipitation isotopic composition**

Due to some gaps in the daily precipitation samples collected at Dome C (described in Sect. 2.3), the timeseries of isotopic composition cannot be used as a direct input to the SISG model. Instead, we generate three artificial timeseries based on (1) the atmospheric temperature, (2) the precipitation-weighted mean annual isotope cycle in precipitation and (3) the arithmetic mean annual isotope cycle in precipitation. A comparison of these three artificial timeseries and the daily observations of the precipitation isotopic composition is shown in Fig. S2 (Supplement S3.1).

The first and second SISG experiment uses the artificial timeseries based on the atmospheric temperature. The precipitation isotopic composition is calculated from the atmospheric temperature using the linear relationships between $\delta^{18}O$ and $\delta D$ of the precipitation samples and the 3 m daily mean temperature (Eq. 1 and 2 in Sect. 3.3.3). Deuterium excess is then calculated from the theoretical values of $\delta^{18}O$ and $\delta D$.

The third SISG experiment uses an artificial timeseries where all days in each month have the same isotopic composition than the corresponding monthly precipitation-weighted mean isotopic composition calculated over five years (results presented in Sect. 3.3.2, Fig. 6c and f).



The fourth experiment uses an artificial timeseries where all days in each month have the same isotopic composition than the corresponding monthly arithmetic mean isotopic composition calculated over five years (results presented in Sect. 3.3.2, Fig. 6b and e).

The fifth experiment uses the daily precipitation isotopic composition modelled by ECHAM6-wiso. In the simulations, all data points outside of the 5-year average ± three standard deviation range were discarded (six data points for $\delta^{18}$O and 20 data points for d-excess over five years). A comparison of the observed daily precipitation isotopic composition and ECHAM6-wiso simulations is shown in Fig. 6 (Sect. 3.3.2).

## 3 Results

### 3.1 Surface moisture fluxes

The water vapor fluxes from the surface to the lower atmosphere for the period 2018-2020 estimated using the meteorological parameters measured at Dome C and the bulk method (described in Sect. 2.4.2) are shown in Fig. 2.

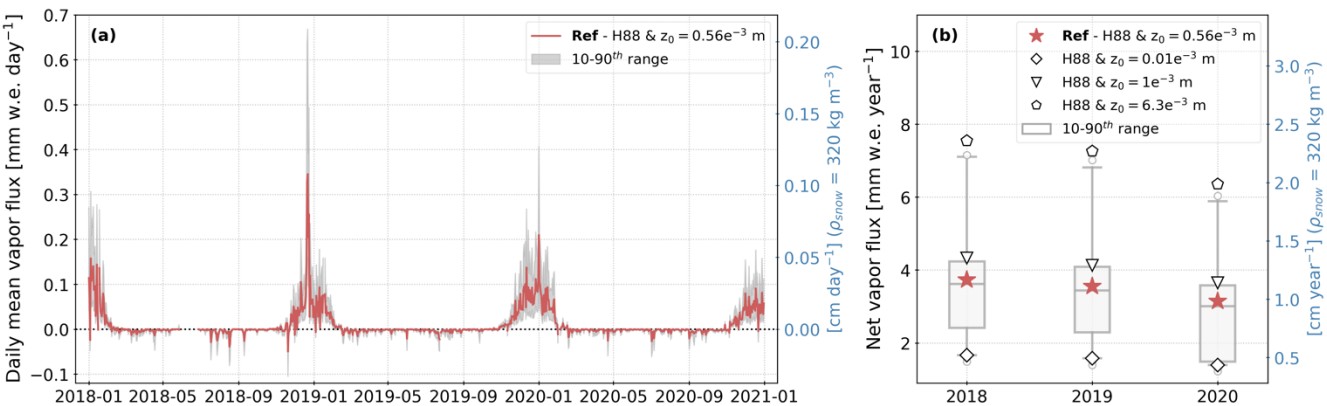

**Figure 2.** Water vapor fluxes at Dome C during the period 2018-2020 (positive for sublimation, negative for condensation). Panel (a) displays the 3-year timeseries of daily mean water vapor flux in mm water equivalent. The red line corresponds to the reference parametrization and the grey shading corresponds to the 10-90$^{th}$ percentile range of all parametrizations (described in Sect. 2.4.2). Panel (b) shows the net annual water vapor flux (sum of sublimation and condensation over one year). The red stars indicate the reference parametrization, and the three other black markers indicate the results using the stability function H88 with different $z_0$. The whiskers of the grey boxplots indicate the 10-90$^{th}$ percentile range of all parametrizations, and the grey circles indicate the outliers outside of this range. The secondary axis to the right in blue gives water vapor fluxes in cm of snow, using a snow density of 320 kg m$^{-3}$ to convert SWE to snow height.

During the 3-year period 2018-2020, the daily mean water vapor flux calculated with the reference parametrization varied from -0.05 (condensation) to 0.35 (sublimation) mm water equivalent per day (red line in Fig. 2a). The seasonality of moisture fluxes over this period is characterised by sublimation during the summer months while little condensation is observed during the rest of the year. This seasonal pattern is observed independently of the parametrization used in the bulk method, which only affects the magnitude of the fluxes (grey shading in Fig. 2a).



The net annual water vapor flux between 2018 and 2020 is positive, meaning a net annual sublimation of snow, regardless of
which parametrization is used (grey boxplots in Fig. 2b). In 2018, 2019, and 2020, water vapor fluxes calculated with the
reference parametrization led to a net mass loss of 3.7, 3.6, and 3.1 mm w.e., respectively (red stars in Fig. 2b), which is
slightly higher than the net water vapor flux of 2.8 mm w.e. in 2015 (Genthon et al., 2017). These values correspond to 1.2,
1.1, and 1.0 cm of snow, respectively, using a snow density of 320 kg m$^{-3}$ to convert from SWE to snow height. They are
doubled when using a roughness length for momentum of $6.3 \times 10^{-3}$ m instead of $0.56 \times 10^{-3}$ m (black pentagons in Fig. 2b)
and divided by approximately two when using a roughness length for momentum of $0.01 \times 10^{-3}$ m (black triangles in Fig.
2b). The net annual water vapor fluxes are increased by 0.1 mm w.e. using a sensor height of 2 m above the surface instead
of 3 m to consider height changes of the sensors (Sect. 2.4.1). During the summer periods only (from November to February,
both months included), water vapor fluxes led to a net sublimation of 4.4, 4.1, and 3.8 mm w.e. in 2018, 2019, and 2020,
respectively (not shown).

**3.2 Snow isotopic composition**

**3.2.1 Temporal variations over five years**

The five-year timeseries of the snow surface and subsurface isotopic composition ($\delta^{18}$O and d-excess) is displayed in Fig. 3,
together with the respective mean annual cycle (monthly means) for each layer over the whole period.

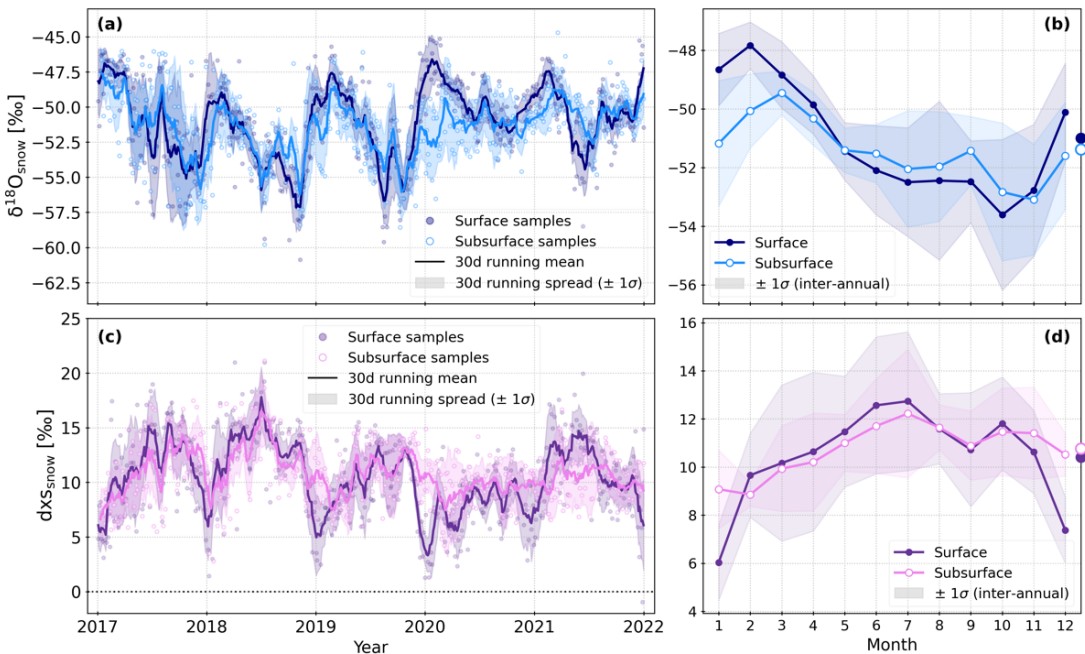

**Figure 3.** Observations of the surface and subsurface snow isotopic composition at Dome C. Panels (a) and (c) show the 5-year timeseries
(2017-2021) for $\delta^{18}$O and d-excess (dxs) respectively. The circles represent the horizontal average between the two samples taken at the
two locations 50 m apart on the sampling lines (described in Sect. 2.2) and the solid lines show a 30-day running mean of these horizontal
averages. The shaded area represents the 30-day running average of the spatial spread between the two samples taken at the two locations



50 m apart on the sampling lines (one standard deviation). Panels (b) and (d) show the mean annual cycles (monthly means) of the surface and subsurface snow layers calculated over the period 2017-2021 for $\delta^{18}O$ and d-excess respectively. The shaded area represents the inter-annual variability around the mean annual cycle (one standard deviation). The symbols on the right vertical axis indicate the mean isotopic composition of the snow surface and subsurface layers across all years. In all four panels, the darker colours correspond to the surface samples and the lighter colours to the subsurface samples.

Considering all samples collected during the five-year period, the $\delta^{18}O$ in the snow has a large amplitude, with values ranging from -60.9‰ to -45.1‰ in the surface layer and from -59.8‰ to -44.7‰ in the subsurface layer (circles in Fig. 3a). Both snow layers have a higher $\delta^{18}O$ during the summertime and lower values in the wintertime (lines in Fig. 3a).

This seasonality is further visible in their respective mean annual cycles (Fig. 3b), with the surface snow $\delta^{18}O$ highest in February (-47.8‰) and lowest in October (-53.6‰, dark blue in Fig. 3b). Compared to the surface snow, the mean annual cycle in the subsurface snow has a smaller amplitude and is shifted in time, with a maximum in March (-49.5‰) and a minimum in November (-53.1‰, light blue in Fig. 3b).

The temporal variation of the surface snow $\delta^{18}O$ is characterized by sharp increases during the summertime followed by slow decreases through the winter, which is particularly clear for the summers 2017-2018, 2018-2019 and 2019-2020 (dark blue line in Fig. 3a). This asymmetric seasonal pattern has been previously identified by Casado et al. (2018) for earlier years and is reflected in the mean annual cycle of the surface snow $\delta^{18}O$ (dark blue in Fig. 3b). A similar pattern is visible in the subsurface snow, although with a reduced amplitude (light blue in Fig. 3a and b).

The variations of $\delta^{18}O$ in the surface and the subsurface snow generally follow each other, except for specific periods when the surface and subsurface differ by several per mill, for example at the beginning of 2020 (solid lines and shaded areas in Fig. 3a). This difference between the two snow layers is reflected in their respective mean annual cycles and is the largest during the summer (Fig. 3b).

The overall mean $\delta^{18}O$ of the surface and subsurface snow is -51.0 ± 0.2‰ and -51.4 ± 0.1‰, respectively (dark and light blue dots in Fig. 3b). The uncertainty around these mean values corresponds to the standard error of the mean (SEM), calculated using the effective number of independent samples in the timeseries (Bretherton et al., 1999, Supplement S2).

As for $\delta^{18}O$, the surface and subsurface snow show large variations in d-excess over the five-year period (Fig. 3c). Considering all samples collected over the period, d-excess ranges from -0.9‰ to 21.0‰ in the surface snow and from 2.8‰ to 21.1‰ in the subsurface snow (dots in Fig. 3c). In opposition to $\delta^{18}O$, high d-excess values are encountered in the wintertime and lower d-excess in the summertime (lines in Fig. 3c).

This seasonality in d-excess is further reflected in the mean annual cycles of both snow layers (Fig. 3d). The surface snow d-excess is the highest in July (12.7‰) and the lowest in January (6.0‰, dark purple in Fig. 3d). As for $\delta^{18}O$, the mean annual cycle in the subsurface snow has also a smaller amplitude compared to the surface layer, with a maximum in July (12.2‰) and a minimum in February (8.9‰, violet in Fig. 3d). However, contrary to $\delta^{18}O$, the mean annual cycle in the subsurface snow does not show a clear time lag compared to the surface. Instead, the variations in the subsurface follows the ones of the surface, apart from the summer months of January and December where the subsurface has larger d-excess than the surface





layer (Fig. 3d). This summertime difference in d-excess between the surface and subsurface layers is also visible in the timeseries, in particular during the summer 2019-2020 (Fig. 3c).

Lastly, the variations in d-excess in the surface snow do not have the same pattern as for $\delta^{18}O$ (sharp increase in summertime and slow decrease in the wintertime). Instead, the surface snow d-excess shows a more symmetric seasonal evolution than $\delta^{18}O$ (dark purple line in Fig. 3c). This symmetry is reflected in the mean annual cycle in d-excess of the surface snow (Fig.

3d).

The overall mean d-excess of the surface and subsurface snow is 10.4 ± 0.2‰ and 10.8 ± 0.1‰, respectively (dark purple and violet dots in Fig. 3d). The uncertainty around these mean values corresponds to the SEM, calculated the same way as for $\delta^{18}O$.

### 3.2.2 Vertical difference between the surface and subsurface snow

In the previous section, we identified a seasonal pattern in the vertical difference between the surface and subsurface snow isotopic compositions ($\delta^{18}O$ and d-excess). The vertical difference is defined here as the isotopic composition of the surface layer minus the isotopic composition of the subsurface layer.

Considering all samples collected during the five-year period (displayed as dots in Fig. 3a and c), the minimum difference in $\delta^{18}O$ between the surface and the subsurface snow is -7.5‰ (surface depleted in $\delta^{18}O$ compared to subsurface) and the

maximum difference is 9.1‰ (surface enriched in $\delta^{18}O$ compared to subsurface). The corresponding minimum and maximum values in the vertical difference in d-excess are -10.6‰ and 12.3‰, respectively.

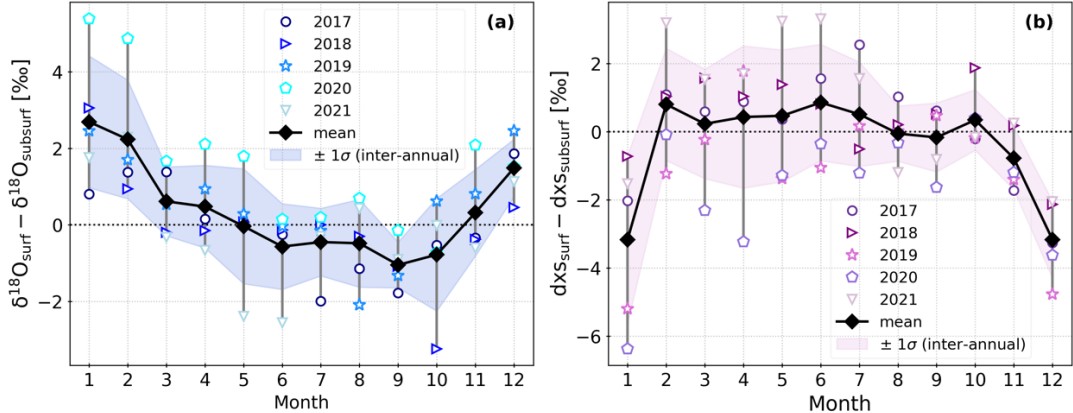

**Figure 4.** Mean annual cycle (monthly means) of the observed vertical difference between the snow surface and subsurface isotopic composition at Dome C. Panel (a) shows the mean annual cycle of the vertical difference in $\delta^{18}O$, panel (b) shows the mean annual cycle

of the vertical difference in d-excess. In both panels, the black diamonds correspond to the overall monthly means calculated over the 2017-2021 period. The shaded area shows the inter-annual variability of individual years around the overall monthly means (one standard deviation) and the coloured markers show the monthly means for each individual year.

The mean annual cycles of the vertical difference in $\delta^{18}O$ and d-excess between the two depths show a clear seasonal pattern

(Fig. 4a and b). The surface snow is relatively enriched in $\delta^{18}O$ from November to April and relatively depleted from May to



October compared to the subsurface (black diamonds in Fig. 4a). The maximum vertical difference is 2.7‰ and occurs in January, the minimum difference is -1.0‰ and occurs in September. However, compared to the inter-annual variability within the averaging period (coloured markers and shaded area in Fig. 4a), only the months of January, February, September, and December show a substantial vertical difference in $\delta^{18}O$ between the snow layers.

In opposition to $\delta^{18}O$, the surface snow has a lower d-excess in the summer months of January and December, with a maximum monthly mean difference of -3.2‰ in January (black diamonds in Fig. 4b). The minimum difference of 0.9‰ is found in June and is, however, negligible compared to the inter-annual variability within the averaging period (coloured markers and shaded area in Fig. 4b).

### 3.3 Precipitation amounts and isotopic composition in observations, ECHAM6-wiso and ERA5 reanalyses

### 3.3.1 Precipitation amounts

To evaluate whether ERA5 reanalysis data and ECHAM6-wiso results correctly capture the precipitation amounts at Dome C, we compare them with the observations in Fig. 5 (all three timeseries scaled to the observed mean annual accumulation, Sect. 2.5).

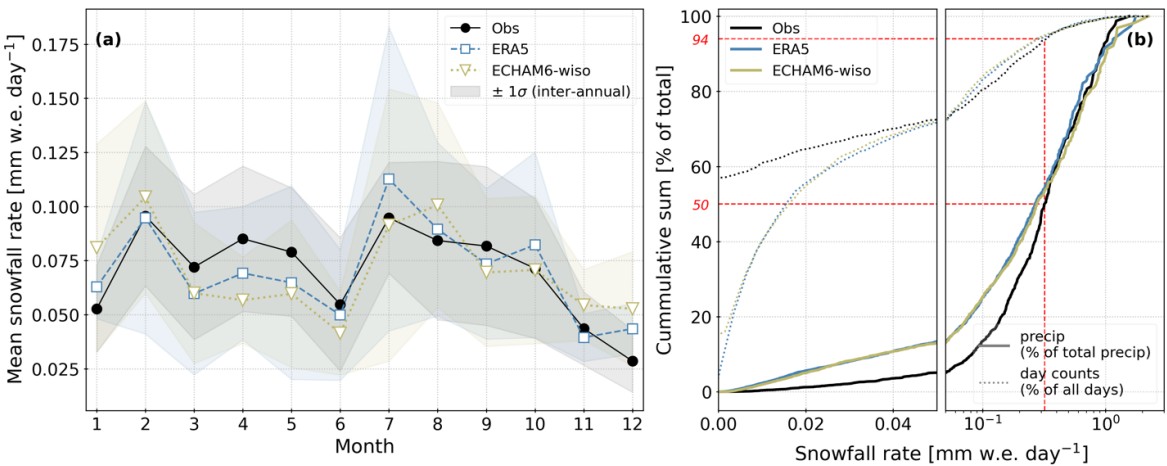

**Figure 5.** Precipitation amounts at Dome C from observations, ERA5 reanalyses and ECHAM6-wiso simulation outputs. Panel (a) shows the mean annual cycles (monthly means) of the daily precipitation amounts (in mm w.e. per day) calculated over the period 2017-2021. Panel (b) shows the cumulative sum of precipitation over 5 years, reported as percentage of the total amount of precipitation (black and coloured thick lines), and the cumulative sum of days (black and coloured dotted lines) against the daily snowfall rate in mm w.e. per day. The red dashed lines guide the reading of the plot to get the snowfall rate and percentage of days responsible for 50% of the total
snowfalls. Note the linear x-axis between 0 and 0.05 mm w.e. day$^{-1}$ and logarithmic above.

Over the whole 5-year period, the precipitation amounts in the observations, ERA5 and ECHAM6-wiso have a comparable seasonal amplitude: from 0.03 to 0.1 mm w.e. day$^{-1}$ for the observations (black in Fig. 5a), from 0.04 to 0.11 mm w.e. day$^{-1}$ for ERA5 (blue in Fig. 5a) and from 0.04 to 0.1 mm w.e. day$^{-1}$ for ECHAM6-wiso (light green in Fig. 5a). The three
precipitation timeseries show a very similar seasonality, with an increase in snowfall rate at the end of the summertime (from




January to February) and in the middle of the winter (from June to July), as well as a decrease from July to December (Fig. 5a).

All three precipitation cumulative sums given by the observations, ERA5 and ECHAM6-wiso show a similar shape with a faster increase with increasing snowfall rates (plain lines in Fig. 5b). The cumulative sums from ERA5 and ECHAM6-wiso

are superposed in the whole range of snowfall rates, whereas the precipitation cumulative sum is lower for the observations up to 0.35 mm w.e. day$^{-1}$, where the three curves meet.

In the observations, 50% of the total snowfalls over five years is due to precipitation events with a snowfall rate above 0.32 mm w.e. day$^{-1}$ (red dashed lines and black plain line in Fig. 5b), which represent only 6% of all days within the 5-year period (red dashed lines and black dotted line in Fig. 5b). It should be noted that these results are dependent on the precipitation

samples collected on site, which have biases (i.e. too little precipitation to collect or blown snow instead of true precipitation, Sect. 2.3). Similarly, in ERA5 precipitation, 50% of the total snowfalls is due to precipitation events with snowfall rates above 0.27 mm w.e. day$^{-1}$ which corresponds to 7% of all days (Fig. 5b). In ECHAM6-wiso simulations, 50% of the total snowfalls is due to precipitation events with snowfall rates above 0.28 mm w.e. day$^{-1}$ which corresponds to 6% of all days (Fig. 5b).

For all three timeseries (observations, ERA5 and ECHAM6-wiso), we find that the largest precipitation events described above (contributing to 50% of the total accumulation) occur alongside higher temperatures than average. In the observations, the mean temperature during all precipitation days within the 5-year period and with snowfall rates above 0.32 mm w.e. per day is 2.8°C warmer than the mean temperature over the whole period (-52.1°C). For ERA5, the mean temperature (given by ERA5) during the largest precipitation events with snowfall rates above 0.27 mm w.e. per day is 7.7°C above the mean

temperature of -49.5°C. Lastly, for ECHAM6-wiso, the mean temperature (given by the model) during the largest precipitation events with snowfall rates above 0.28 mm w.e. per day is 5.7°C above the mean temperature of -51.1°C. These results are in agreement with previous studies (Kino et al., 2021; Servettaz et al., 2023).

### 3.3.2 Precipitation isotopic composition

The daily temporal variability of the precipitation isotopic composition ($\delta^{18}$O and d-excess) from both observations and

ECHAM6-wiso simulations is presented in Fig. 6, together with the corresponding mean annual cycles over the same period. In this section, all mean values across the whole period are given with an uncertainty corresponding to the SEM (see Sect. 3.2.1).

The observed precipitation $\delta^{18}$O shows a large seasonal cycle, ranging from -82.6‰ to -21.8‰, with higher values in the summertime and lower values in the wintertime (dark blue dots in Fig. 6a), following the atmospheric temperature (grey line

in Fig. 6a). The mean value over the whole period is -56.2 ± 0.5‰ (dark blue dot in Fig. 6b). In comparison, the daily precipitation $\delta^{18}$O modelled by ECHAM6-wiso show a similar seasonality as the observations, with higher and lower $\delta^{18}$O in the summertime and wintertime respectively, and a comparable amplitude with values ranging from -82.9‰ to -22.8‰ (blue triangles in Fig. 6a). However, the mean modelled $\delta^{18}$O in precipitation over the whole period is higher than the observed



one (-52.7 ± 0.5‰, blue triangle in Fig. 6b). The precipitation-weighted overall means are higher than the arithmetic means,
for both observations and ECHAM6-wiso simulations (-53.4 ± 0.5‰ and -45.7 ± 0.5‰, respectively, dark blue dot and blue
triangle in Fig. 6c).

The observed mean annual cycle (arithmetic monthly means) in the precipitation $\delta^{18}O$ is characterised by the highest $\delta^{18}O$ in
December (-46.5‰) and the lowest $\delta^{18}O$ in June (-62.0‰, dark blue dots in Fig. 6b). In comparison, the modelled mean
annual cycle shows a similar seasonality but systematically higher than the observed one, ranging from -59.9‰ in June to -
38.4‰ in December (blue triangles in Fig. 6b). The difference between the observations and ECHAM6-wiso results is
especially large in the summertime (November to January), with a maximum difference of 8.1‰ in December.

The observed and modelled precipitation-weighted $\delta^{18}O$ mean annual cycles also show a similar seasonality (Fig. 6c).
However, the bias between ECHAM6-wiso and observations is visible throughout the whole year, with a maximum
difference of 10.2‰ in October (Fig. 6c). The large difference between the observations and ECHAM6-wiso during the
summertime (arithmetic mean, Fig. 6b) decreases when the precipitation $\delta^{18}O$ is weighted by the precipitation amounts (Fig.
6c).

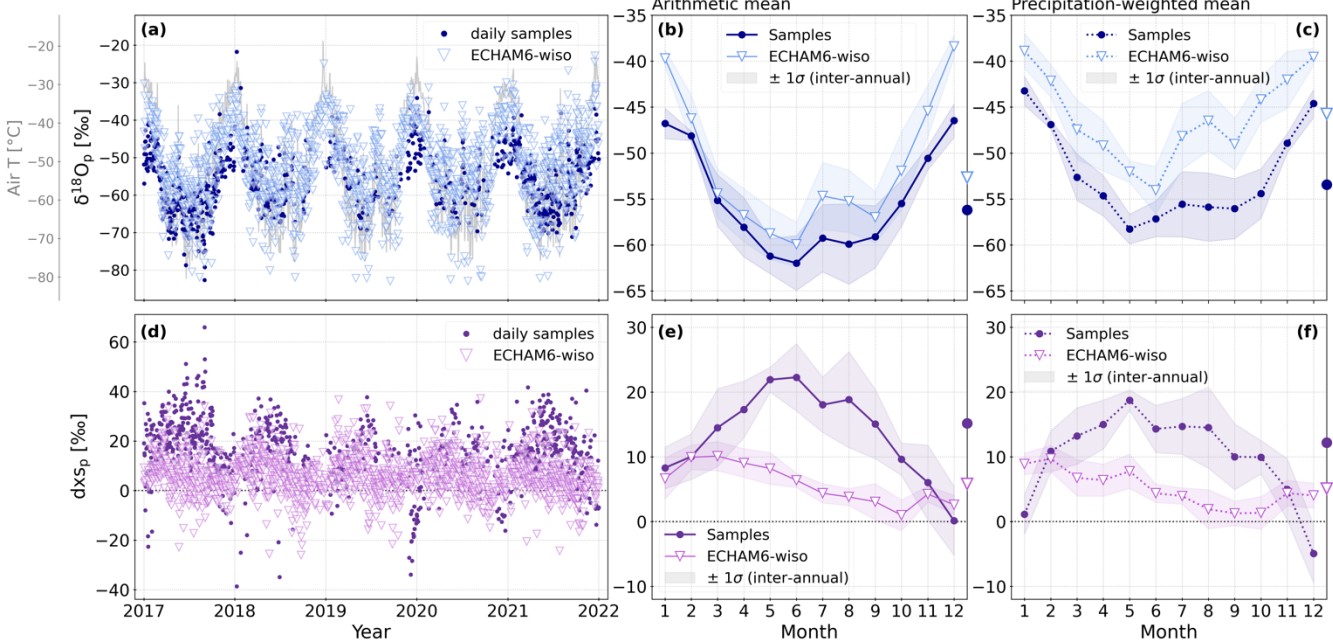

**Figure 6.** Observed and modelled precipitation isotopic composition at Dome C. Panels (a) and (d) show the 5-year time series (2017-
2021) for $\delta^{18}O$ and d-excess (dxs) respectively. The dots represent the daily samples collected in the field (sample description in Sect. 2.3)
and the triangles represent the daily precipitation isotopic composition modelled by ECHAM6-wiso (description of simulations in Sect.
2.5). In grey in panel (a) is displayed the atmospheric temperature measured at 3 m (described in Sect. 2.4.1). Panels (b) and (e) show the
observed (dots) and modelled by ECHAM6-wiso (triangles) mean annual cycle, calculated over the period 2017-2021 (arithmetic means)
for $\delta^{18}O$ and d-excess, respectively. The shaded area represents the inter-annual variability around the mean annual cycle (one standard
deviation). The markers on the right vertical axis indicate the mean observed and simulated precipitation isotopic composition over the
whole period. Panels (c) and (f) are the same as (b) and (e) but showing the weighted monthly means (by precipitation amounts).





The d-excess in the precipitation samples also shows a large seasonal amplitude, with values ranging from -38.6‰ to 65.9‰ (dark purple dots in Fig. 6d). d-excess is in anti-phase to $\delta^{18}$O, with maximum values found in the wintertime and minimum values in the summertime, which have been previously identified by Stenni et al. (2016) and Dreossi et al. (2023). The mean

observed precipitation d-excess over the whole period is equal to 15.2 ± 0.5‰ (dark purple dot in Fig. 6e). In comparison, the daily precipitation d-excess modelled by ECHAM6-wiso has a lower amplitude with values ranging from -26.0‰ to 37.2‰ (violet triangles in Fig. 6d). The overall mean modelled d-excess is also lower than the observations with a value of 5.8 ± 0.3‰ (violet triangle in Fig. 6e). The precipitation-weighted overall means are lower than the arithmetic means, for both observations and ECHAM6-wiso results (12.2 ± 0.5‰ and 5.1 ± 0.3‰, respectively, dark purple dot and violet triangle

in Fig. 6f).

The observed mean annual cycle (arithmetic monthly means) in the precipitation d-excess is characterised by the lowest value in December (0.1‰) and the highest value in June (22.3‰, dark purple dots in Fig. 6e). In comparison, the mean annual cycle in the modelled precipitation d-excess has a lower amplitude than the observed one and a different timing in the minimum and maximum (from 1.0‰ in October to 10.1‰ in March, violet triangles in Fig. 6e). As opposed to $\delta^{18}$O, the

difference between the observations and ECHAM6-wiso simulations is large in the wintertime (April to October), with a maximum difference of 15.9‰ in June.

Compared to the arithmetic mean annual cycles for d-excess, the bias between the precipitation-weighted monthly means in ECHAM6-wiso and the observations is reduced in the winter but increased in the summer months January and December (Fig. 6f). The maximum difference is also lowered to 12.6‰ and found later in the year (August, Fig. 6f).


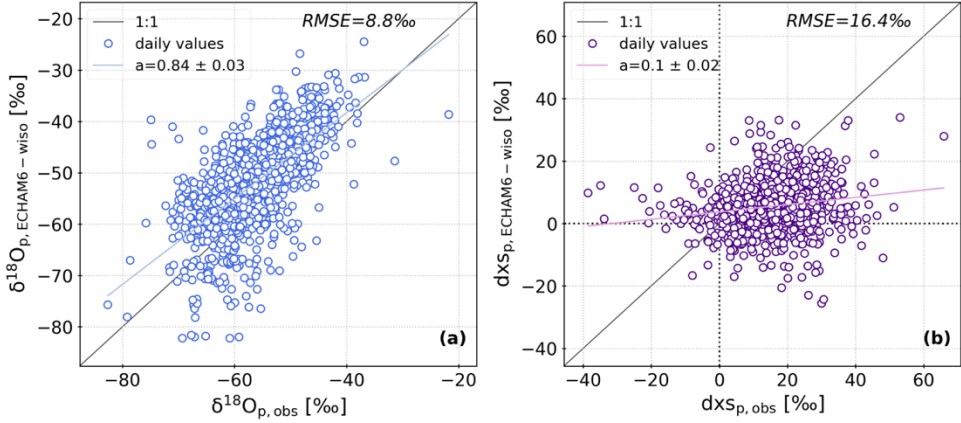

**Figure 7.** Daily observed precipitation $\delta^{18}$O (a) and d-excess (b) versus daily modelled precipitation isotopic composition at Dome C. All precipitation samples collected between 2017 and 2021 are shown (described in Sect. 2.3), together with the corresponding daily values from ECHAM6-wiso simulation results (described in Sect. 2.5). The coloured lines correspond to the linear fits between the observations

and the model results (slope coefficients given in legends, both linear regression coefficients are significant with a p-value < 0.001).

We evaluate the performance of ECHAM6-wiso to model the observed daily precipitation isotopic composition in Fig. 7. The daily precipitation $\delta^{18}$O modelled by ECHAM6-wiso shows a good agreement with the observations, with a linear



regression slope of 0.84 ± 0.03 (1.0 being the perfect fit) and a root mean square error (RMSE) of 8.8‰ (Fig. 7a). The model
mostly overestimates the observations with an increase in bias towards more depleted values (Fig. 7a). For d-excess, the
ECHAM6-wiso model results only poorly represent the observations, with a linear regression slope of 0.1 ± 0.02 and a
RMSE of 16.4‰ (Fig. 7b). These results are similar to the comparison of the precipitation samples and ECHAM6-wiso
simulation results between 2008 and 2017 (RMSE=6.1‰ for $\delta^{18}$O and 13.6‰ for d-excess, Dreossi et al., 2023).

### 3.3.3 Isotope versus temperature relationships in precipitation

From the isotopic composition of the daily precipitation samples collected on site and the corresponding daily average
temperature measured at 3 m above the surface (described in Sect. 2.4.1), we determine the following linear relationships
over the 2017-2021 time-period:

$$\delta^{18}O_p = 0.47 \pm 0.01 \times T_{3m} - 31.2 \pm 0.6 \text{ ‰} \tag{1}$$

$$\delta D_p = 3.3 \pm 0.1 \times T_{3m} - 262 \pm 4 \text{ ‰} \tag{2}$$

Equation (1) has a coefficient of determination ($R^2$) of 0.62 and Eq. (2) has a $R^2$ of 0.63. Both linear regression slopes are
significant (p-values < 0.001, Fig. A1). There are no significant changes in the linear relationships if using the daily mean
atmospheric temperature of the day before the sample collection day instead of the daily mean temperature of the sampling
day itself (Fig. A1). The slopes in Eq. (1) and (2) are also comparable to the ones found for the 2008-2010 period (0.49 ±
0.02 for $\delta^{18}$O, Stenni et al., 2016) and for the 2008-2017 period (0.52 ± 0.01 for $\delta^{18}$O, 3.52 ± 0.07 for $\delta$D, Dreossi et al.,
525   2023).

In ECHAM6-wiso simulations, the linear relationships between the modelled precipitation isotopic composition and the
modelled temperature differ from the observed ones:

$$\delta^{18}O_{p, \text{ECHAM6wiso}} = 0.68 \pm 0.02 \times T_{2m, \text{ECHAM6wiso}} - 17.0 \pm 0.9 \text{ ‰} \tag{3}$$

$$\delta D_{p, \text{ECHAM6wiso}} = 5.3 \pm 0.1 \times T_{2m, \text{ECHAM6wiso}} - 134 \pm 6 \text{ ‰} \tag{4}$$

These relationships are also established over the period from 2017 to 2021. Equation (3) has a $R^2$ of 0.54 and Eq. (4) has a $R^2$
of 0.57. Both linear regression slopes are significant (p-values < 0.001, Fig. A2).

## 4 Discussion

### 4.1 Contribution of precipitation to the snow isotopic composition

In this section we discuss the contribution of the precipitation isotopic composition to the intra-monthly and seasonal
variability in the snow $\delta^{18}$O and d-excess at Dome C, by comparing the observations and the results from the SISG
experiments (described in Sect. 2.5).



### 4.1.1 Surface snow

Some of the features observed in the surface snow $\delta^{18}O$ variability are reproduced by the different experiments performed with the SISG model (Fig. 8a). For example, the observed summer $\delta^{18}O$ values in the surface snow are correctly reproduced

in four out of five experiments ("iso from T & cst accu", "iso from T & ERA5 accu", "iso from wg. mm & obs accu", "iso from ar. mm & obs accu"). All five experiments correctly reproduce the seasonality in $\delta^{18}O$ observed in the surface snow, with higher values in the summer and lower values in the winter (Fig. 8a and b). In addition, the two model experiments "iso from T & ERA5 accu" and "iso from ECHAM6 & ECHAM6 accu" reproduce some of the short-term increases in the surface snow $\delta^{18}O$ during the wintertime, such as the events in August 2018, July 2020, or August 2021 (orange shadings in

Fig. 8a). The other three experiments fail to reproduce these events because of the inputs given to the SISG model: either a constant daily precipitation that weighs equally the days with high or low $\delta^{18}O$ ("iso from T & cst accu") or a daily isotopic composition that does not vary within one month and fails to represent the high $\delta^{18}O$ events ("iso from wg. mm & obs accu" and "iso from ar. mm & obs accu", Fig S2 in Supplement S3.1).

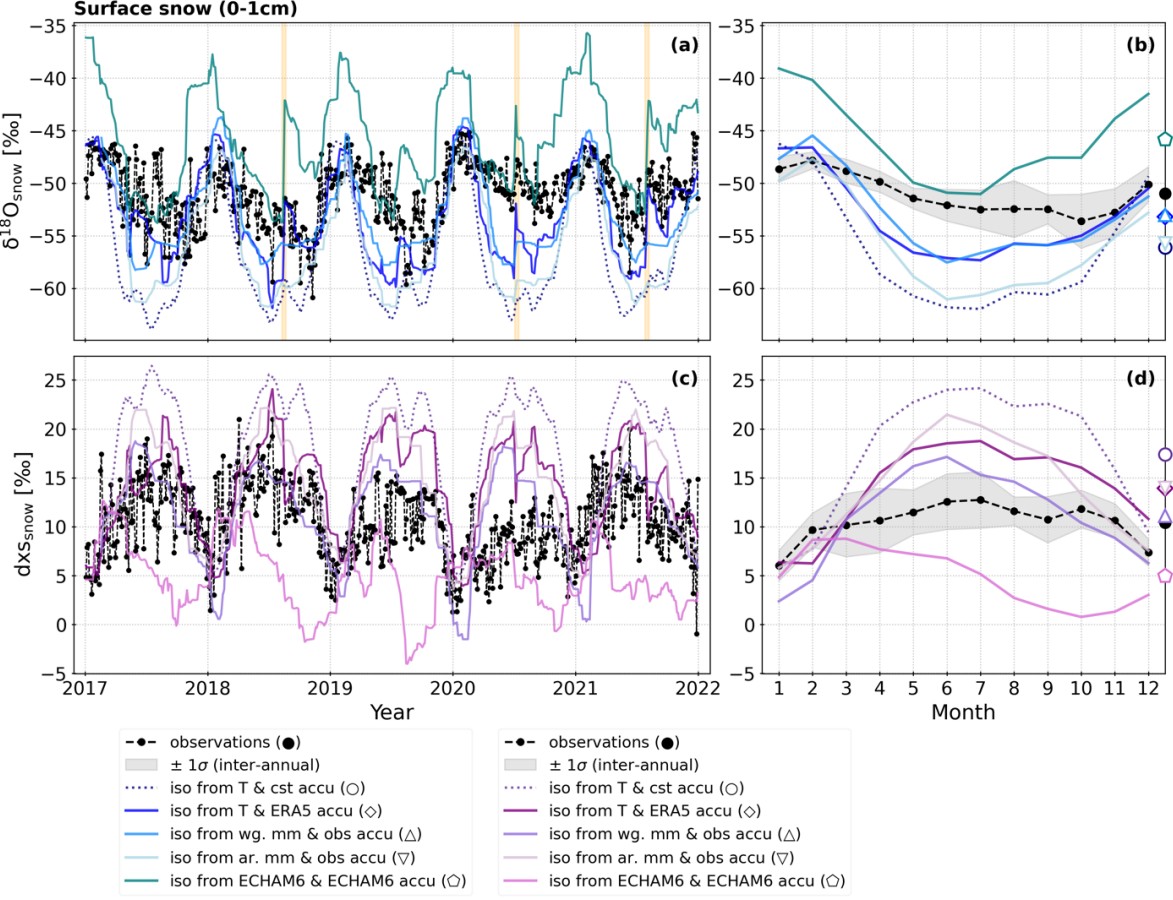

**Figure 8.** Comparison between observed and simulated surface snow (0 to 1 cm depth) isotopic composition at Dome C. Panels (a) and (c) show the 5-year time series (2017-2021) of $\delta^{18}O$ and d-excess (dxs) respectively. The black dots represent the horizontal average between



the two samples taken at the two locations 50 m apart on the sampling lines (described in Sect. 2.2, also presented in Fig. 3). The coloured lines (dotted and plain) represent the different SISG model experiments (described in Table 3). In panel (a), the orange vertical shadings highlight three specific events (August 2018, July 2020, August 2021). Panels (b) and (d) show the mean annual cycles (monthly means) of $\delta^{18}O$ and d-excess, respectively, calculated over the period 2017-2021. The black dots and line represent the observations and the shaded area represents the inter-annual variability around the mean annual cycle (one standard deviation). The coloured lines (dotted and plain) represent the different SISG experiments (described in Table 3). The markers on the right vertical axis indicate the mean isotopic composition of the observed and simulated snow surface over 5 years (see Table S2 in Supplement S3.2).


However, regardless of the model experiment, the amplitude of the seasonal cycle in the surface snow $\delta^{18}O$ is systematically overestimated compared to the observations (Fig. 8a and b). All five experiments fail to reproduce the slow decrease in $\delta^{18}O$ during the wintertime observed in the surface snow. This is particularly visible for the winters 2019 and 2020 where all simulated surface snow has too low $\delta^{18}O$ values (Fig. 8a). Most of the short-term variations in the snow $\delta^{18}O$ occurring within the month are also not reproduced by any of the experiments (Fig. 8a). In addition, the experiment "iso from

ECHAM6 & ECHAM6 accu" gives too high $\delta^{18}O$ compared to the observations throughout the whole time-period (blue-green lines in Fig. 8a and b), which is consistent with the positive bias between the precipitation-weighted $\delta^{18}O$ modelled by ECHAM6-wiso and the observations, identified in Sect. 3.3.2 (Fig. 6c).

As for $\delta^{18}O$, all model experiments fail to reproduce the variability in d-excess observed in the surface snow. Although in some of the experiments ("iso from T & cst accu", "iso from T & ERA5 accu", "iso from wg. mm & obs accu", "iso from ar.

mm & obs accu"), the low summer d-excess in the surface snow is well captured (Fig. 8c), all the simulated mean annual cycles have a too large amplitude (Fig. 8d). The experiment "iso from ECHAM6 & ECHAM6 accu" provides worse results than the other experiments (pink line in Fig. 8c and d), which can be explained by the poor representation of the observed precipitation-weighted annual cycle in d-excess by ECHAM6-wiso simulations, identified in Sect. 3.3.2 (Fig. 6f).

To evaluate which SISG experiment best represent the observed snow surface isotopic composition, we perform a linear regression between all modelled and observed monthly means $\delta^{18}O$ and d-excess in the five-year period. The linear regression slopes (a) and the RMSE between the model results and the observations are summarized in Table 4.

The best representation of the observed $\delta^{18}O$ in the surface snow is given by the experiment "iso from wg. mm & obs accu" (a=1.1, RMSE=3.4‰, Table 4). The experiment "iso from T & cst accu" results in the highest RMSE of all configurations

tested (RMSE=6.9‰, Table 4), which is expected due to the intermittent nature of precipitation at Dome C. In addition, the experiment "iso from ECHAM6 & ECHAM6 accu" gives a high RMSE (6.3‰, Table 4).

For d-excess, out of the five model experiments, "iso from wg. mm & obs accu" gives the lowest RMSE (3.9‰, Table 4). As for $\delta^{18}O$, the experiment "iso from T & cst accu" results in the highest RMSE (8.8‰, Table 4). Lastly, the experiment "iso from ECHAM6 & ECHAM6 accu" has a RMSE of 7.1‰ and a non-significant slope compared to the observations (p-value

> 0.05).

**Table 4.** Linear regression slope (a) and RMSE between the observed and modelled monthly mean isotopic composition of the surface layer ($\delta^{18}O$ in normal font, d-excess in parenthesis and italic). The sample size is 60 for all experiments except "iso from wg. mm & obs





accu" and "iso from ar. mm & obs accu" (n=59). All linear slopes are significant (p-value < 0.05), except the ones marked with an asterisk
(*, p-value > 0.05).

| Experiment | "Iso from T & cst accu" | "Iso from T & ERA5 accu" | "Iso from wg. mm & obs accu" | "Iso from ar. mm & obs accu" | "Iso from ECHAM6 & ECHAM6 accu" |
|---|---|---|---|---|---|
| a | 1.4 *(1.4)* | 1.2 *(0.9)* | 1.1 *(0.9)* | 1.3 *(1.1)* | 1.1 *(0*)* |
| RMSE (‰) | 6.9 *(8.8)* | 3.7 *(5.3)* | 3.4 *(3.9)* | 5.8 *(5.5)* | 6.3 *(7.1)* |

### 4.1.2 Subsurface snow

For $\delta^{18}O$ in the subsurface snow, we find comparable results as for the surface layer. We find that all five model experiments
fail to capture the short-term variations in the subsurface snow, and all experiments result in a too-large amplitude in the
mean annual cycle over five years (Fig. 9a and b). The systematic positive bias given by the model experiment "iso from
ECHAM6 & ECHAM6 accu" is also visible in the simulated subsurface layer (Fig. 9a and b). In addition, all five model
experiments provide a mean annual cycle shifted compared to the observed one, with a maximum $\delta^{18}O$ occurring one to two
months later than the observed maximum (Fig. 9b).

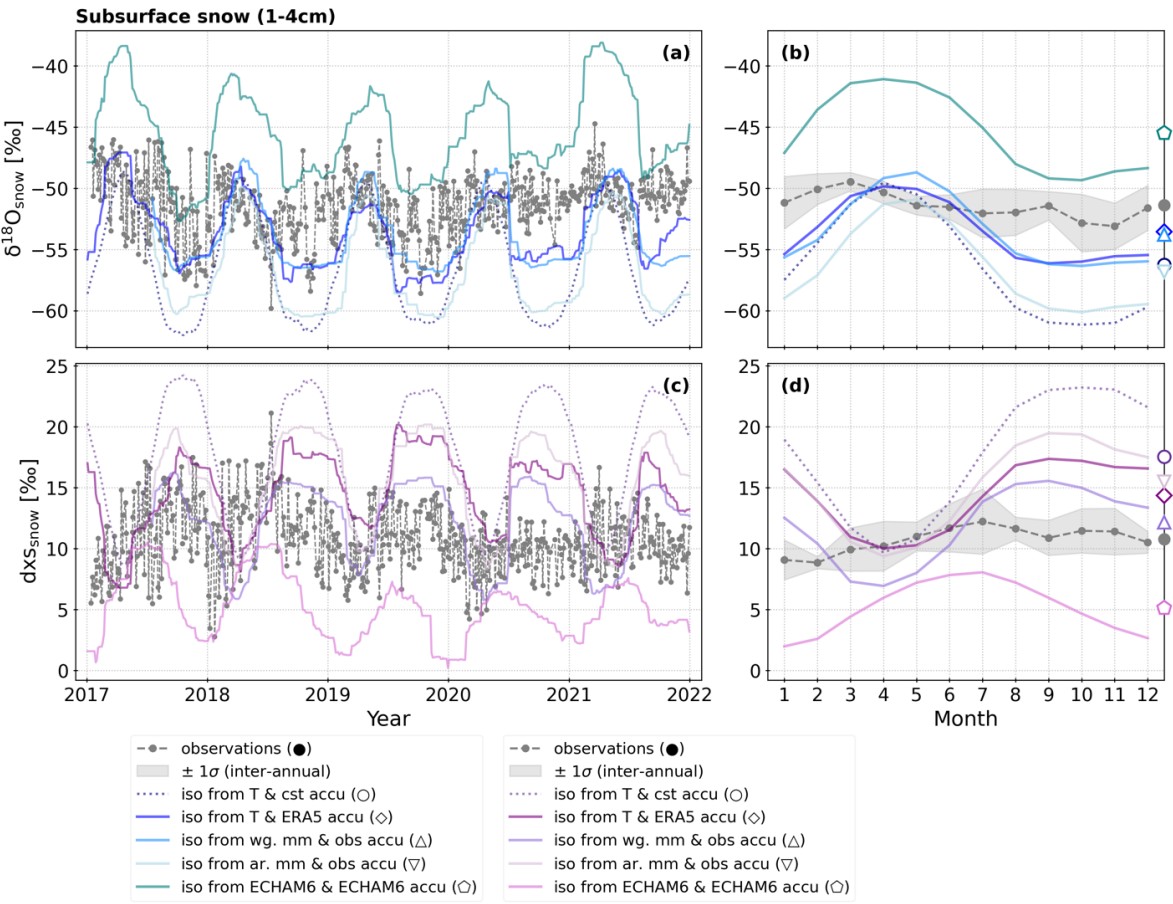

**Figure 9.** Comparison between observed and simulated subsurface snow (1 to 4 cm depth) isotopic composition at Dome C. Panels (a) and
(c) show the 5-year time series (2017-2021) of $\delta^{18}O$ and d-excess (dxs) respectively. The grey dots represent the horizontal average





between the two samples taken at the two locations 50 m apart on the sampling lines (described in Sect. 2.2, also presented in Fig. 3). The coloured lines (dotted and plain) represent the different SISG model experiments (described in Table 3). Panels (b) and (d) show the mean annual cycles (monthly means) of $\delta^{18}O$ and d-excess, respectively, calculated over the period 2017-2021. The grey dots and line represent

the observations and the shaded area represents the inter-annual variability around the mean annual cycle (one standard deviation). The coloured lines (dotted and plain) represent the different SISG experiments (described in Table 3). The markers on the right vertical axis indicate the mean isotopic composition of the observed and simulated snow subsurface over 5 years (see Table S2 in Supplement S3.2).

For d-excess, we find that four out of five of the model experiments ("iso from T & cst accu", "iso from T & ERA5 accu",

"iso from wg. mm & obs accu", "iso from ar. mean & obs accu") give too high d-excess variations over time, which is reflected in the too large amplitude of the modelled mean annual cycle compared to the observed one (Fig. 9c and d). As for $\delta^{18}O$, the annual cycle in d-excess modelled in these three experiments is shifted, with the minimum in d-excess found two months later than the observed one (Fig. 9d).

On the other hand, although giving too low d-excess compared to the observations, the experiment "iso from ECHAM6 &

ECHAM6 accu" results in a similar mean annual cycle as in the observations, with minimum d-excess in the summertime and maximum d-excess during the wintertime (pink line in Fig. 9d). However, because the precipitation d-excess modelled by ECHAM6-wiso poorly represent the observations (Fig. 6e and f), we argue that the good resemblance (besides the negative bias) between the modelled and observed d-excess in the subsurface snow is due to compensating effects between the precipitation d-excess and precipitation amounts.


As for the surface snow, we perform a linear regression between all modelled and observed monthly means $\delta^{18}O$ and d-excess. The linear slopes (a) and the RMSE between the model results and the observations are summarized in Table 5.

Contrary to the surface layer, the best representation of $\delta^{18}O$ in the subsurface layer is given by the experiment "iso from T & ERA5 accu" (a=0.8, RMSE=3.2‰, Table 5). The model experiment "iso from ECHAM6 & ECHAM6 accu" gives the

best correlation with observations, although with the highest RMSE (a=1.1, RMSE=6.7‰, Table 5).

For d-excess in the subsurface snow, only the experiment "iso from ECHAM6 & ECHAM6 accu" gives a significant slope with the observations (a=0.8, RMSE=6.0‰, Table 5) while all other experiments have a non-significant linear slope with the observations (p-value > 0.05, Table 5). However, as stated above, we argue that this good agreement between the simulated subsurface snow in the experiment "iso from ECHAM6 & ECHAM6 accu" is due to compensating effects between the

precipitation d-excess and precipitation amounts.

**Table 5.** Linear regression slope (a) and RMSE between the observed and modelled monthly mean isotopic composition of the subsurface layer ($\delta^{18}O$ in normal font, d-excess in parenthesis and italic). The sample size is 60 for all experiments except "iso from wg. mm & obs accu" and "iso from ar. mm & obs accu" (n=54). All linear slopes are significant (p-value < 0.05), except the ones marked with an asterisk

(*, p-value > 0.05).

| Experiment | "Iso from T & cst accu" | "Iso from T & ERA5 accu" | "Iso from wg. mm & obs accu" | "Iso from ar. mm & obs accu" | "Iso from ECHAM6 & ECHAM6 accu" |
|---|---|---|---|---|---|
| a | 1.0 *(0.5\*)* | 0.8 *(0.2\*)* | 0.5 *(0.4\*)* | 0.7 *(0.3\*)* | 1.1 *(0.8)* |
| RMSE (‰) | 6.2 *(8.4)* | 3.2 *(5.1)* | 3.5 *(3.5)* | 6.1 *(6.0)* | 6.7 *(6.0)* |



### 4.1.3 SISG setup and inputs

In the Sect. 4.1.1 and 4.1.2, we showed that none of the five SISG experiments fully represent the variability observed in the snow surface and subsurface isotopic composition ($\delta^{18}O$ and d-excess). However, the model was designed with a fixed depth
for both snow samples to best represent the observations, although the depths of the samples collected in the field might have varied over time (Sect. 2.2). Changing the sample depths in the SISG model does not significantly improve the agreement between the model results and the observations (Table S3 in Supplement S3.3).

The inputs of the model might also explain some of the discrepancies between the observed and modelled snow layers. For example, there are fewer small precipitation events in the observations than given by ERA5 and ECHAM6-wiso (Fig. 5b),
which could lead to an underestimation of the contribution of smaller precipitation events (e.g. diamond dust) to the total accumulation and explain the differences between the experiments "iso from wg. mm and obs accu" and "iso from ar. mm and obs accu" and the observations (Fig. 8 and 9). Some snowfall events in ERA5 and ECHAM6-wiso could also be occurring too early or too late and explain why the short-term increases and decreases in the snow isotopic composition are not reproduced in the experiments using ERA5 and ECHAM6-wiso precipitation amounts (Fig. 8a and c, Fig. 9a and c). In
addition, we scaled up all three precipitation timeseries to match the mean annual accumulation of 8 cm year$^{-1}$ at Dome C (Sect. 2.5), although this value can vary and estimations over other time periods have given higher mean accumulation rates (e.g. Frezzotti et al., 2005). Lastly, in three out of five experiments, we used an artificial daily precipitation isotopic composition (Sect. 2.5), but none of these represent accurately the observed precipitation isotopic composition (Fig. S2 in Supplement S3.1).

### 4.2 Transfer of the isotopic signal between precipitation, surface, and subsurface snow

In this section we discuss the potential processes at the surface of the ice sheet responsible for the discrepancies between the precipitation signal and the observed snow surface and subsurface layers and assess the overall impact of these processes on the transfer of the isotopic signal between precipitation to the snow.

### 4.2.1 Short-term, inter-annual and seasonal variability in the snow isotopic composition

Several studies have shown that water vapor fluxes between the lower atmosphere and the snow can modify the isotopic composition of the snow surface (Casado et al., 2021b; Hughes et al., 2021; Wahl et al., 2021, 2022) and deeper firn cores (Dietrich et al., 2023). Although the impact depends on the isotopic composition of the atmospheric water vapor above, sublimation generally leads to an enrichment in $\delta^{18}O$ of the snow surface and a lowering of its d-excess. In addition, Casado et al. (2018) hypothesized that the inter-annual variability of the summer snow surface isotopic composition is related to the
strength of metamorphism and surface vapor fluxes.

We showed that at Dome C, sublimation occurs during the summertime (Fig. 2a) and that there is a net annual sublimation of snow (Fig. 2b), which means that the surface snow isotopic composition must have been affected by these fluxes. In the



snow samples collected at the surface and just below, we observe some inter-annual variability in the vertical difference between the surface and subsurface snow isotopic compositions (Fig. 4). This difference between the layer exposed to the atmosphere and the subsurface could be explained by the inter-annual variability in vapor fluxes (Fig. 2a). For example, the snow surface is most enriched in $\delta^{18}O$ compared to the subsurface in December 2019 and January-February 2020 (Fig. 4). In parallel, we observe strong sublimation occurring in November and December 2019 (Fig. 2a), probably enriching the snow surface in $\delta^{18}O$, and leading to a much more enriched surface snow compared to the subsurface in the samples collected in the following months. Yet, the highest sublimation rate occurring in December 2018 (Fig. 2a) is not necessarily reflected in the vertical difference between the surface and subsurface snow layers (Fig. 4). This could be explained by the length of the sublimation period, critical for the overall impact on the snow surface (Hughes et al., 2021; Wahl et al., 2022).

The mean seasonal cycle in the difference between the surface and subsurface isotopic compositions can, on the other hand, most likely be explained by the depths of the samples that integrate the precipitation events fallen during the last couple of months for the surface sample and up to the previous winter for the subsurface sample (assuming no removal or redistribution and no compaction of snow, Fig. S3 in Supplement S3.4). The higher $\delta^{18}O$ and lower d-excess in the summer precipitation compared to the winter precipitation (Fig. 6) explains why the mean summer surface snow is enriched in $\delta^{18}O$ and has a lower d-excess than the subsurface snow (Fig. 4).

If we would include vapor fluxes in the simple SISG model, the amplitude of the modelled seasonal cycle in the surface snow would be increased (summer months enriched in $\delta^{18}O$ and reduced in d-excess), increasing the discrepancy with the observations (Fig. 8b and d). This shows that an additional process reducing the amplitude of the mean annual cycle in the snow surface takes place. Diffusion is a good candidate.

The isotopic composition of the snowpack is affected by diffusion of water molecules along isotopic and temperature gradients (Johnsen et al., 2000; Gkinis et al., 2014). The addition of this process in our model would possibly reduce the too large amplitude of the seasonal cycles in the synthetic snow layers to match the observations (Fig. 8b and, 9b and d). In addition, some of the short-term variations observed in the subsurface snow that are not explained by the incoming precipitation (Fig. 9a and c) could instead be explained by changes occurring in the surface snow and diffused downwards, as suggested by Casado et al., (2018).

Lastly, although Dome C is not affected by strong katabatic winds, surface winds can still be strong enough to erode and redistribute the snow (Libois et al., 2014). In a study of the process of snow accumulation at Dome C, Picard et al. (2019) showed that the surface snow changes very frequently due to snow erosion and accumulation by small patches (10% of the surface only is affected by each precipitation events), and that the snow in a patch can be as old as one year at the surface, while another patch was deposited by the last precipitation event. In the snow samples collected at Dome C, we observe higher differences in $\delta^{18}O$ and d-excess between two consecutive sampling days of the surface and subsurface snow when the wind speed increases from one sampling day to another (Fig. S4 in Supplement S4), as well as a decrease in $\delta^{18}O$ and d-excess vertical difference between the surface and the subsurface snow (Fig. S5 in Supplement S4). This could indicate that



some of the short-term variations observed in the snow surface isotopic composition that cannot be explained by new precipitation falling on the surface (Fig. 3, 8a and c) be caused instead by wind erosion and redistribution.

The wind blowing at the surface of the ice sheet during or beside snowfall could lead to a mixing of new precipitation with already deposited snow. This process was proposed by Casado et al. (2018) to explain the slow decrease in the surface snow $\delta^{18}$O during the winter, a pattern observed in our study (Fig. 3a and b) and that cannot be explained by precipitation only (Fig. 8a and b).


Overall, our results show evidence that the snow isotopic composition is affected by post-depositional processes at different timescales. Disentangling the contribution of the different processes described above on the final isotopic signal found in the snow is beyond the scope of this study and requires the use of an isotope-equipped snowpack model that includes post-depositional processes at the surface, such as the one developed recently by Wahl et al. (2022) and Dietrich et al. (2023).


### 4.2.2 Mean effect of post-depositional processes

The datasets presented in this study permit to quantify the overall impact of post-depositional processes on the snow isotopic composition. Table 6 summarizes the mean isotopic composition over five years of the precipitation (weighted average by the observed precipitation amounts, see Sect. 2.5.1), the snow surface, and the subsurface snow. The mean isotopic composition of precipitation excluding all samples with negative d-excess and modelled by ECHAM6-wiso are also provided. All mean values are given with their respective standard error (see Sect. 3.2.1 and Supplement S2) and 95% confidence interval (Student t-test, Supplement S2).


**Table 6.** Five-year mean isotopic composition of precipitation (weighted by the observed precipitation amounts) and snow samples ($\delta^{18}$O in normal font, d-excess in parenthesis and italic). The mean values are given with their respective standard error (see Sect. 3.2.1) and in brackets their 95% confidence interval (Student t-test, Supplement S2).


| | Observations | Observations (excl. samples with dxs < 0) | ECHAM6-wiso |
|---|---|---|---|
| Precipitation-weighted (‰) | -53.4 ± 0.5 [1.0] *(12.2 ± 0.5 [1.0])* | -54.3 ± 0.4 [0.9] *(13.9 ± 0.4 [0.8])* | -45.7 ± 0.5 [0.9] *(5.1 ± 0.3 [0.5])* |
| Surface snow (‰) | -51.0 ± 0.2 [0.4] *(10.4 ± 0.2 [0.5])* | – – | – – |
| Subsurface snow (‰) | -51.4 ± 0.1 [0.3] *(10.8 ± 0.1 [0.3])* | – – | – – |

We express the mean effect of post-depositional processes at Dome C as the difference between the 5-year mean observed precipitation-weighted isotopic composition of precipitation and the 5-year mean isotopic composition of the surface snow. Considering all precipitation and snow samples, the surface snow $\delta^{18}$O is 2.4‰ higher than in precipitation and the surface snow d-excess is 1.8‰ lower than in precipitation (Table 6). However, to exclude any imprint of sublimation on the precipitation isotopic composition, we re-compute the mean isotopic composition of precipitation discarding all the samples with a d-excess below zero, as in Steen-Larsen et al. (2011). Although the zero threshold might be arbitrary, it is supported by laboratory and field studies that showed a decrease in d-excess during snow sublimation and metamorphism (Hughes et





al., 2021; Casado et al., 2021b; Harris Stuart et al., 2023). In addition, Stenni et al. (2016) and Dreossi et al. (2023) already stated that some of the samples collected in the summertime at Dome C might have been affected by sublimation and we observe in the precipitation dataset presented here that the samples with very negative d-excess were collected during the summertime (e.g. December 2019, Fig. 6d) when sublimation occurred (Fig. 2a). We therefore argue that removing all

precipitation samples with a d-excess below zero improves the representation of the true precipitation falling at Dome C. Now considering this new precipitation timeseries, the $\delta^{18}O$ in the surface snow is 3.3‰ higher than in precipitation and the d-excess in the surface snow is 3.5‰ lower than in precipitation (Table 6). Both differences are significant on a 95% confidence level (Table 6).

The weighted mean isotopic composition of precipitation is computed using the observed precipitation amounts, estimated

from the weights of the samples collected on site (Sect. 2.3 and 2.5.1). However, as discussed in Sect. 4.1.3, these amounts might not accurately represent the precipitation falling at Dome C. Using the precipitation amounts given by ERA5 instead, the weighted mean $\delta^{18}O$ of precipitation is -57.6‰ and 18.0‰ for d-excess, which leads to larger differences between the precipitation and the snow surface: $\delta^{18}O$ in the surface snow is 6.6‰ higher than in precipitation and d-excess in the surface snow is 7.6‰ lower than in precipitation.

The discrepancies between the precipitation and the surface snow show that the mean isotopic composition of precipitation is not preserved from snowfall to the surface snow. On average over a 5-year period at Dome C, post-depositional processes lead to an enrichment in $\delta^{18}O$ of the snow surface by 3.3‰ to 6.6‰ (depending on which precipitation amounts are considered) and a lowering of d-excess in the snow surface by 3.5‰ to 7.6‰ compared to the precipitation signal. It is still to be determined the individual contribution of the different post-depositional processes (discussed in Sect. 4.2.1) on the

isotopic difference observed between the precipitation and the snow surface.

In contrast to the isotopic difference between the precipitation and surface snow, there is no significant difference between the mean surface and subsurface snow isotopic compositions, for both $\delta^{18}O$ and d-excess (Table 6). This shows that despite a seasonality in the vertical difference between the two snow layers (discussed in Sect. 4.2.1), the mean isotopic composition of the surface snow layer is preserved in the top few centimetres of the snowpack.

From the sampling of a 2 m snow pit at Dome C, Touzeau et al. (2016) reported mean values (plus-minus standard error of the mean) of -51.1 ± 0.2‰ and 9.1 ± 0.2‰ for the $\delta^{18}O$ and d-excess profiles, respectively. As found in our study for the upper layers of the snow, the average isotopic composition of the snow pit is enriched in $\delta^{18}O$ and has a lower d-excess than the mean incoming precipitation isotope signal.

The weighted-mean isotopic composition of precipitation modelled by ECHAM6-wiso is 7.7‰ higher in $\delta^{18}O$ and 7.1‰ lower in d-excess than the observed precipitation (Table 6, Fig. 6c and f). These large differences result from the biases identified between the modelled and observed precipitation isotopic composition (Sect. 3.3.2, Fig. 6 and 7), and show the limitations of using ECHAM6-wiso simulations to interpret the isotopic composition of the snow at Dome C. A thorough investigation of the biases in ECHAM6-wiso is beyond the scope of this study, but might arise from different processes in





the model, such as the environmental conditions at the moisture source region, the moisture transport and pathway, the
supersaturation parametrization or the condensation height and temperature above Dome C.

### 4.2.3 Seasonal $\delta^{18}$O versus temperature relationships in precipitation and snow

In precipitation, the $\delta^{18}$O-temperature relationship is determined between the $\delta^{18}$O in the daily precipitation samples and the
corresponding daily atmospheric temperature (Sect. 3.3.3, Eq. (1)). For the surface and subsurface snow, we determine the
relationships between the $\delta^{18}$O composition of each snow sample and the weighted-average temperature (by precipitation
amounts) over the entire period during which precipitation accumulated and formed the snow sample (Fig. A3). We use the
observed precipitation amounts to determine this averaging period (Fig. S3 in Supplement S3.4). Table 7 summarizes the
linear regression slopes (a) with their standard errors and the coefficient of determination ($R^2$).

**Table 7.** Summary of mean seasonal $\delta^{18}$O versus temperature linear relationships in precipitation and snow samples between 2017 and
2021. The slopes (a) are given with their standard error and all three slopes are significant (p-value < 0.001).

|  | Precipitation | Snow surface | Snow subsurface |
|---|---|---|---|
| a (‰ °C$^{-1}$) | 0.47 ± 0.01 | 0.17 ± 0.01 | 0.12 ± 0.02 |
| $R^2$ | 0.62 | 0.25 | 0.1 |

We find a mean seasonal $\delta^{18}$O-temperature slope in precipitation of 0.47 ± 0.01‰ °C$^{-1}$, in agreement with previous studies
for the same location for earlier years (Stenni et al., 2016; Dreossi et al., 2023). We find a mean seasonal slope of 0.17 ±
0.01‰ °C$^{-1}$ in the surface snow (top 1 cm) and a mean seasonal slope of 0.12 ± 0.02‰ °C$^{-1}$ in the subsurface snow (1 to 4
cm, Table 7). The surface snow slope coincides with the one found for the snow top 1-2 mm at Dome C (0.14 ± 0.03‰ °C$^{-1}$,
Touzeau et al., 2016) but is lower than the ones found by Casado et al. (2018) for similar samples in earlier years (slopes
ranging from 0.22 to 0.49‰ °C$^{-1}$). This difference can be due to uncertainties on the precipitation amounts used to determine
the averaging period of temperature (i.e. some precipitation days with colder temperatures are not captured), the method used
to determine the averaging period of temperature (i.e. assumes that snowfall is not redistributed or removed and does not
include snow compaction, Supplement S3.4), or due to the different method used in Casado et al. (2018) where they compute
the slope as the ratio of the maximum amplitude in $\delta^{18}$O and the maximum amplitude in temperature.

Our results indicate that deriving a temporal slope from seasonal variations of $\delta^{18}$O in the surface snow to be used for ice
core studies is not consistent with other in-situ and proxy temperature reconstructions. First, the surface snow $\delta^{18}$O is not
showing a symmetric seasonal cycle. Instead, the $\delta^{18}$O minimum is shifted towards spring (Fig. 3, Sect. 3.2.1) while the
minimum temperature occurs in winter. Secondly, our determination of the seasonal relationship between the snow $\delta^{18}$O and
the atmospheric temperature results in a weaker correlation and a lower slope (0.12 – 0.17‰ °C$^{-1}$, Table 7) than what is
found for precipitation (0.47‰ °C$^{-1}$, Table 7). Using the $\delta^{18}$O-temperature slope found in the upper layers of the snow for
temperature reconstructions from ice cores would lead to unrealistic glacial-interglacial temperature change exceeding 30°C.
The change in the seasonal $\delta^{18}$O-temperature relationship between the precipitation and the upper layers of the snow shows





the difficulty to interpret quantitatively the $\delta^{18}$O variations in the snow at Dome C in terms of temperature. Instead, it provides the opportunity to document the effects of post-depositional processes at this site and improve the future quantitative interpretation of water isotopic records in ice cores from Dome C for longer timescales (decadal and longer

timescales).

### 4.3 Long-term perspectives for the interpretation of ice core isotope records

The classical paleothermometer approach (Lorius and Merlivat, 1975) to determine past temperature variations from the water isotopic profiles in ice cores uses the present-day spatial slope between the water isotopic composition of the surface snow and the local temperature. However, as illustrated in Casado et al. (2017), this spatial slope differs from the various

estimates of the temporal slope between temperature and $\delta^{18}$O found in the literature and in this study, determined for different regions and timescales. These discrepancies show the need of calibration of the isotopic paleothermometer. Such calibration has been done using alternative methods at the glacial-interglacial scale (e.g. borehole thermometry and firn properties, Buizert et al., 2021), or using isotope-enabled GCMs to determine the relationship between temperature and water isotopes during past periods (e.g. Werner et al., 2018). However, in these models, the absence of explicit modelling of

how the water isotopic signal is archived in the snow and firn limits their use for paleo-reconstructions.

To progress towards an accurate quantitative interpretation of isotopes in ice cores, we recommend developing a *proxy system model* that can be coupled to isotope-enabled GCMs. Such a model has been developed recently by Wahl et al. (2022) and Dietrich et al. (2023) for Greenland and includes mechanical processes leading to the recording of isotopes at the ice sheet's surface and in ice cores. The datasets presented in our study can therefore be useful to calibrate and validate this kind

of *proxy system model* at intra-annual, seasonal, and inter-annual timescales at the deep drilling site of the EPICA Dome C ice core.

### 5 Conclusions

We have presented the compilation of new and existing datasets of the isotopic composition of precipitation, snow surface and subsurface together with meteorological parameters, reanalysis products, model outputs and a simple modelling

approach to investigate the origin of the stable water isotopic signal in the upper layers of the snowpack at Dome C, East Antarctica.

From in-situ meteorological observations, we have quantified the amount of water vapor fluxes between the snow and the lower atmosphere. Our results show that vapor fluxes contribute to the surface mass balance at Dome C with a net annual mass loss of snow from 3.1 to 3.7 mm w.e. year$^{-1}$ between 2018 and 2020, which corresponds to 12 to 15% of the annual

surface mass balance. Sublimation is relatively strong in the summertime and there is small condensation in the wintertime.

From the comparison between the snow surface and subsurface isotopic compositions and a simple modelling approach, we show that the precipitation deposited onto the ice sheet cannot explain the variability observed in the snow isotopic



composition ($\delta^{18}$O and d-excess) at seasonal to intra-monthly timescales, highlighting the existence of post-depositional processes affecting the snow isotopic composition.

We quantified the cumulative effect of post-depositional processes on the surface snow over five years by comparing the mean isotopic compositions of the precipitation and the snow layers. Our results show that post-depositional processes lead to an enrichment of the snow surface (top 1 cm) in $\delta^{18}$O by 3.3‰ to 6.6‰ (depending on the precipitation amounts considered) and a lowering of the surface snow d-excess by 3.5‰ to 7.6‰. In addition, our results show that the mean isotopic composition of the subsurface snow (1 to 4 cm depth) is not significantly different than the surface snow, which

indicates that the mean isotopic composition of the surface snow layer is preserved in the top centimetres of the snowpack and that the processes altering the precipitation isotopic signal mainly take place in the top one centimetre of the snow.

In both the observations, ERA5 reanalyses and ECHAM6-wiso simulation, we showed that 50% of the accumulation at Dome C over five years occurs during large but rare precipitation events associated with warmer temperatures than average, leading to a warm bias in the $\delta^{18}$O record of precipitation compared to the mean annual temperature. We further find

different seasonal relationships between the atmospheric temperature and $\delta^{18}$O in the precipitation and the snow, showing the difficulty to interpret the variations of $\delta^{18}$O in the snow at Dome C.

Overall, our results show that post-depositional processes at the ice sheet's surface have an impact on the isotopic signal (both $\delta^{18}$O and d-excess) found in the upper layers of the snowpack at Dome C, East Antarctica. The datasets presented in our study open the possibility to calibrate and validate *proxy system models* including post-depositional processes that are

needed to quantitatively attribute the different mechanisms building up the isotopic signal in the surface snow and to accurately reconstruct the climatic information from the water stable isotope records in ice cores.



**Appendix A**

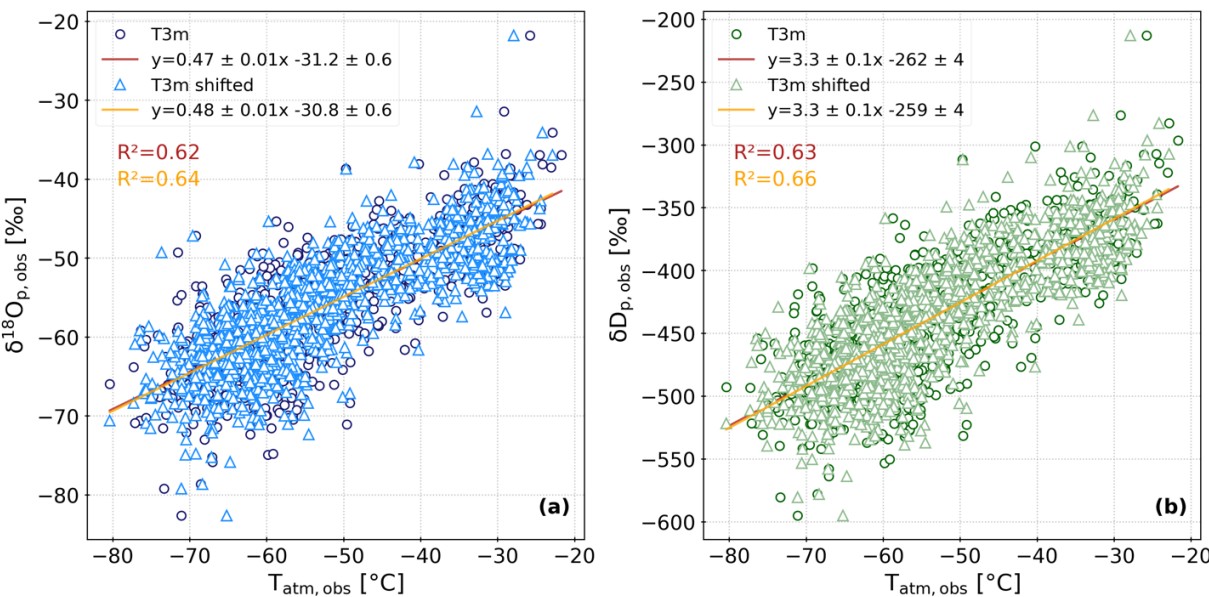

**Figure A1.** Daily observed precipitation $\delta^{18}O$ (a) and $\delta D$ (b) versus observed daily mean temperature at Dome C. All precipitation samples collected between 2017 and 2021 are shown against the corresponding daily mean temperature (T3m, dark colours) and the daily mean temperature of the day before (T3m shifted, light colours). The linear regressions with the associated coefficients of determination ($R^2$) are

shown. Both linear slopes are significant (p-values < 0.001).

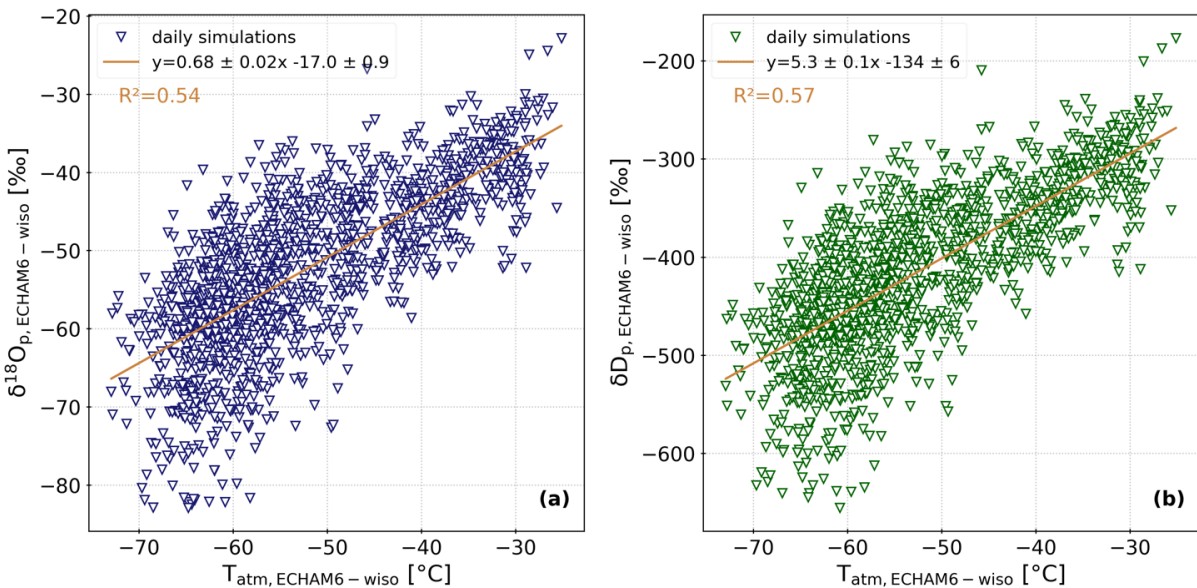

**Figure A2.** Daily precipitation $\delta^{18}O$ (a) and $\delta D$ (b) modelled by ECHAM6-wiso versus daily mean temperature modelled by ECHAM6-wiso. The linear regressions with the associated coefficients of determination ($R^2$) are shown. Both linear slopes are significant (p-values <

0.001).

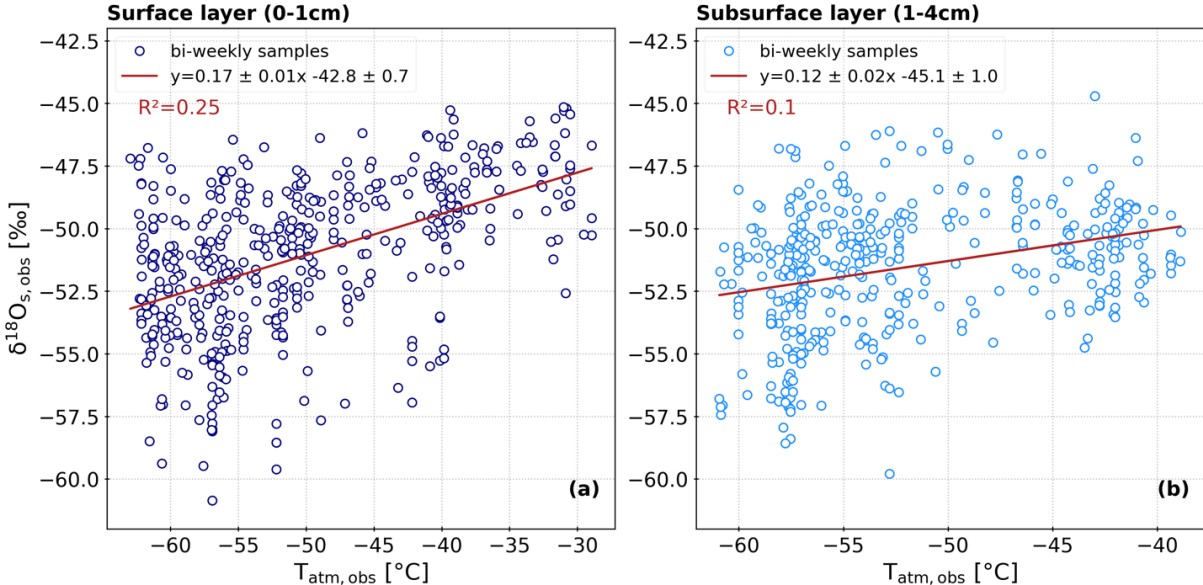

**Figure A3.** Bi-weekly observed snow surface (a) and subsurface (b) $\delta^{18}O$ versus temperature at Dome C. All snow samples collected between 2017 and 2021 are shown against the weighted-average temperature (by precipitation amounts) over the period corresponding to the accumulation of the snow samples (see Sect. 4.2.3). The linear regressions with the associated coefficients of determination ($R^2$) are shown. Both linear slopes are significant (p-values < 0.001).

*Code availability.* Python scripts for the estimations of moisture fluxes with the bulk method and for the SISG model will be uploaded on Zenodo upon acceptance of the article and DOIs will be provided.

*Data availability.* The DOIs for the two datasets in review with PANGAEA will be provided as soon as possible and latest at publication date. The CNR4 dataset will be published upon acceptance of the article.

*Author contribution.* IO, HCSL, BS and AL designed the study and contributed to the analysis. IO performed the formal analysis and visualization. HCSL, BS and AL supervised the study. MC designed the first version of the simple model (SISG) and provided inputs to the study. GP and LA acquired and shared radiation data (CNR4) and coordinated the snow sample collection executed by the winter-over staff. CG acquired meteorological data (USt) and provided inputs to the study. GD and IO performed the isotopic analysis of the precipitation samples. BM performed the isotopic analysis of the snow samples. AC and MW performed the ECHAM6-wiso simulation. IO wrote the first draft of the manuscript and all authors contributed to reviews and edits.

*Competing interests.* The authors declare that they have no conflict of interest.



*Acknowledgements.* This publication was generated in the frame of the DEEPICE project. The project has received funding
from the European Union's Horizon 2020 research and innovation programme under the Marie Sklodowska-Curie grant
agreement no. 955750. Snow samples and radiation data (CNR4) have been collected within the frame of the French Polar
Institute (IPEV) project NIVO 1110. Precipitation samples and its isotopic analysis have been conducted in the framework of
the projects PNRA 2013/AC3.05 (PRE-REC) and PNRA18_00031 (WHETSTONE) of the Italian National Antarctic
Research Program "Programma Nazionale di Ricerche in Antartide" (PNRA) funded by MIUR (now MUR). We
acknowledge using data from the project CALVA 1013 and GLACIOCLIM observatories supported by the French Polar
Institute (IPEV) and the Observatoire des Sciences de l'Univers de Grenoble (OSUG)
(https://web.lmd.jussieu.fr/~cgenthon/SiteCALVA/CalvaData.html) and thank Etienne Vignon from Laboratoire de
Météorologie Dynamique (Paris, France) for his valuable input on the bulk method and vapor fluxes estimations. We also
acknowledge using data from the Italian Antarctic Meteo-Climatological Observatory
(IAMCO, https://www.climantartide.it/) collected in the framework of the PNRA/IPEV 'Routine Meteorological Observation
at Station Concordia' project and from the Baseline Surface Radiation Network (BSRN, https://bsrn.awi.de/). AL has
received funding from the European Research Council under the European Union Horizon 2020 Excellent Science
program (H2020/20192024)/ERC grant agreement no. 817493 (ERC ICORDA). HCSL has received funding from the
European Research Coucil (ERC) under the European Union's Horizon 2020 research and innovation programme: Starting
Grant SNOWISO (grant agreement no. 759526). We also thank the logistics staff and winter-over crews at Concordia station
for all the field sampling and instrumental maintenance during investigated periods and L. Dietrich for proofreading this
article.

*Financial support.* This research has been supported by the European Union's Horizon 2020 research and innovation
programme (grant no. 955750).

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
