# Peer review of "Surface processes and drivers of the snow water stable isotopic composition at Dome C, East Antarctica – a multi-datasets and modelling analysis"

_EGUsphere, 2024_

## Referee Comment (RC2)

Review of "Surface processes and drivers of the snow water stable isotopic composition at Dome C, East Antarctica – a multi-datasets and modelling analysis" by Ollivier et al.

The authors present an extensive analysis of existing and new data of stable water isotopes from precipitation, surface and subsurface snow samples collected next to the Dome C research station in Antarctica over the period from 2017 to 2021. They complement this data with a modelling approach – the Snow Isotopic Signal Generator (SISG) – which tries to simulate the observed isotopic composition with different input parameters. Additionally, they analyse the output from an isotope-enabled GMC (ECHAM6-wiso).

The authors put a lot of effort into combining different isotope data sets from different campaigns. This data is very valuable for the community. I appreciate the dedication of this team to collect and measure these samples and to provide an in-depth analysis to the community. Such data and the knowledge that the community gains from it are a necessary step to better understand the post-depositional processes on ice sheets and the climatic imprint in ice cores.

Specific comments (major)

The manuscript reads lengthy in parts and explains very detailed the observations of precipitation, surface and subsurface snow isotopic composition as well as the modelling approach. In parts it is difficult to follow and to remember all details from the comparisons of different observations and the modelling results. The readability can be improved by shortening parts. Additionally, chapter 4.1 (*Contribution of precipitation to the snow isotopic composition*) shows even more new figures and information. To me, (part of) this section provides many results and does not necessarily discuss them. The authors might consider moving this (or part of this) part to the results section and focus on the comparison of their findings to other studies as well as on implications of their findings in the discussion (as they do from chapter 4.2 onwards).

Additionally, I would like to see comments or a chapter about the limitations of the study. Some limitations are mentioned in some parts, but I am missing a comprehensive analysis of the shortcomings and the reliability of their dataset and, especially, the assumptions on which their study is based on (e.g. isotopes represent local air temperature, scaling of ERA5 precipitation, large surface features and intermittent accumulation at their study site, etc).
- Figure 8, for example, show a large variability between consecutive samples. This can be related to a large spatial heterogeneity in surface features (e.g. Picard et al., 2019), likely causing variability and noise in surface and subsurface snow isotopic compositions. How do you explain these differences? How realistic is it that these two samples, 50m apart, sampled twice a week, represent the local weather/climate, considering the reported surface processes and post-depositional processes (e.g. snow erosion and re-deposition)?

- Sublimation is often discussed as a reason to alter the surface snow isotopic composition significantly throughout the 5-year period. The authors discuss the difference between surface and subsurface snow isotopic composition, but do not use or apply the mentioned model by Wahl et al. (2022) to simply estimate the depth of snow affected by sublimation. Moreover, they mention (l. 695ff.) that the location at Dome C on the East Antarctic Plateau has a very intermittent snow accumulation in time and space (which is well-known). Besides presenting the new datasets, this study does not provide more insights into the mechanism of post-depositional processes affecting the stored climatic information in stable water isotopes in such environments. Extending their simulations with e.g. diffusion and/or sublimation (models for both do already exist, (e.g. Münch et al., 2016; Wahl et al., 2022)) or simply estimating the effect of these processes, would help to quantify the role of these processes and to further disentangle ambient noise, post-depositional processes, and the climatic imprint in stable water isotopes in precipitation as well as in surface and subsurface snow.
- ECHAM6-wiso is compared to the observations and the SISG simulations. What is the benefit of including a GCM here? Can you provide any suggestions for the modelling community how to improve the representation of isotopes in isotope-enabled GCMs? Why not using a smaller model, such as Wahl et al. (2022) or Dietrich et al. (2023) developed when they proved that they can reproduce observations?

Specific comments (minor)

- L. 59: low*er* atmosphere
- L. 81: water stable isotopes -> stable water isotopes
- L. 81f.: It is written "snow surface and subsurface" and later "surface snow and subsurface snow". Please consistently use one term throughout the manuscript.
- L. 118: Is the uncertainty determined with an independent quality control sample?
- L. 118: two standard deviation*s*
- L. 118 and l. 142: First, the uncertainty is given by two standard deviations and later, in l. 142, by one standard deviation of a quality standard. Why are you using different measure for uncertainty? It seems that the first uncertainty is lower when using one standard deviation, especially for dD (0.7 ‰/ 2 = 0.35 ‰). Does this influence the d-excess presented in this study?
- L. 151: The time step is one hour. How is a 1-hour average calculated from a 1-hourly spaced timeseries?
- L. 160: Why are you not averaging the data to 1-hour time steps as well (as the AWSIT data)?
- L. 167 and 169: Which test was used to test for significance?
- L. 191: flux
- Table 1: The sampling rate of the CALVA measurements is given as 30s. Here, you are referring to the used averages. "Measurement time step" is in my opinion not the correct term here. Maybe you can rephrase it.

- L. 287: Which relationship is used here to calculated d$^{18}$O and dD from atmospheric temperature?
- L. 460: How do you explain the fact that the precip-weighted means are higher than then arithmetic means?
- L. 469f.: You write that "the summertime" difference between observations and ECHAM6-wiso "decreases when the precipitation d18O is weighted by the precipitation amounts". To me, this only accounts for December and January while the overall difference increases. Can you explain why? What is your conclusion here?
- Figure 7:
    - Why are you only reporting the RMSE and not the correlation as well?
    - I assume that the data shown in the scatter plot are the same as in Figs. 6a and 6d. It seems that ECHAM6-wiso suggests more days with precipitation than the observations do. How are you handling this? Are you accounting for these or are you ignoring these days?
- L. 528f: Are these the equations used in the ECHAM6-wiso model itself or are these empirically calculated?
- L. 811ff.: My understanding of a proxy system model might be different than yours. Following Evans et al. (2013), for instance, a *proxy system model may be defined as the complete set of forward and mechanistic processes by which the response of a sensor to environmental forcing is recorded and subsequently observed in a material archive.* Wahl et al. (2022) and Dietrich et al. (2023) did a great job in coming up with statistical approaches to e.g. calculate expected isotopic variations in surface and subsurface snow from sublimation. However, I see their models rather as an input to more sophisticated (proxy system) models.

References:

Dietrich, L. J., Steen-Larsen, H. C., Wahl, S., Jones, T. R., Town, M. S., and Werner, M.: Snow-Atmosphere Humidity Exchange at the Ice 940 Sheet Surface Alters Annual Mean Climate Signals in Ice Core Records, Geophysical Research Letters, 50, e2023GL104249, https://doi.org/10.1029/2023GL104249, 2023.

Evans, M. N., Tolwinski-Ward, S. E., Thompson, D. M., & Anchukaitis, K. J.: Applications of proxy system modeling in high resolution paleoclimatology. Quaternary science reviews, 76, 16-28, 2013.

Münch, T., Kipfstuhl, S., Freitag, J., Meyer, H., and Laepple, T.: Regional climate signal vs. local noise: a two-dimensional view of water isotopes in Antarctic firn at Kohnen Station, Dronning Maud Land, Clim. Past, 12, 1565–1581, https://doi.org/10.5194/cp-12-1565-2016, 2016.

Picard, G., Arnaud, L., Caneill, R., Lefebvre, E., and Lamare, M.: Observation of the process of snow accumulation on the Antarctic Plateau by time lapse laser scanning, The Cryosphere, 13, 1983–1999, https://doi.org/10.5194/tc-13-1983-2019, 2019.

Wahl, S., Steen-Larsen, H. C., Hughes, A. G., Dietrich, L. J., Zuhr, A., Behrens, M., Faber, A. -K., and Hörhold, M.: Atmosphere-Snow Exchange Explains Surface Snow Isotope Variability, Geophysical Research Letters, 49, https://doi.org/10.1029/2022GL099529, 2022.

---

## Author Comment (AC1)

Response to anonymous referee #1

We thank Reviewer 1 for their time and effort to provide detailed and constructive feedback on the manuscript, which has improved the quality of the study. We have addressed all comments below and propose to implement the changes in a revised version of the manuscript.

Black: reviewer comment
Blue: author's response
Green: revised text (**bold: new text**)

General comments

The effects of post-depositional processes (water vapor exchanges between surface snow and atmosphere, diffusion of water vapor in the firn, and wind redistribution) on water stable isotopic composition in surface snow in the inland of polar ice sheets strongly limit the quantitative paleo-temperature reconstructions by ice-core stable isotopic records from these regions. In this study, the authors attempted to evaluate the mean effect of post-depositional processes at Dome C using a combination of existing and new dataset of precipitation, surface snow and subsurface isotopic compositions over 2017-2021. It is noted that the sampling schemes are elaborate and the sampling temporal resolution is high, which is not easy to perform at very harsh meteorological conditions such as at Dome C but is very necessary to address the problem of post-depositional processes. I appreciate the authors for their hard work at Dome C and their persistent efforts on this complex issue on the post-depositional processes. Although I agree with the authors for their most interpretations of the observations, I think some explanations should be added and the different post-depositional processes should be clarified.

Specific comments

I note that precipitation, surface snow and subsurface snow show similar $\delta^{18}O$ seasonal cycle, but the maximum (minimum) value of $\delta^{18}O$ occurs in different month, with highest $\delta^{18}O$ value in January and lowest value in May for precipitation, a maximum of $\delta^{18}O$ in February and a minimum in October for surface snow, and a maximum of $\delta^{18}O$ in March and a minimum in November for subsurface snow. I think it must be due to the post-depositional processes, but why the post-depositional processes can lead to the $\delta^{18}O$ shift in time?

We agree with the reviewer that there is a shift in time of the monthly mean maximum $\delta^{18}O$ in the precipitation, the snow surface and the snow subsurface. However, we argue that this is not only due to post-depositional processes but also because of the time period that the samples represent or integrate.

For precipitation, if we consider the arithmetic monthly means, the maximum $\delta^{18}O$ occurs in January and the minimum in June (Fig. 6b, l.462-463). This is explained by the fact that the precipitation $\delta^{18}O$ follows nicely the seasonality of the atmospheric temperature (Fig. 6a), which has a monthly mean maximum in December and minimum in June (Genthon et al., 2021, also valid for the 2017-2021 period). Now considering the precipitation-weighted monthly means, the minimum and maximum $\delta^{18}O$ occur in May and January, respectively, which also relates to the seasonality of atmospheric temperature.

Now the snow surface (here defined as the upper first centimetre of the snowpack) integrates several precipitation events, up to 2 months before the sample collection (Fig. S3). This explains why the highest $\delta^{18}O$ in the snow surface is found in February: the snow surface at this time integrates the precipitation events fallen in December and January with the highest $\delta^{18}O$. It is less clear for the minimum $\delta^{18}O$ in the snow surface (October) because it occurs more than 2 months after the minimum in precipitation (May). This could be due for example to snow erosion and redistribution in winter that mixes winter precipitation with already deposited (and more enriched) snow, as discussed l.703-706.

The same integration effect applies to the snow subsurface (1 to 4 cm deep): its maximum in $\delta^{18}O$ is shifted later in the year compared to the snow surface because it integrates the precipitation events fallen onto the ice sheet up to 6 months prior to sample collection (Fig. S3), so that the high $\delta^{18}O$ summer precipitation events are only "recorded" in the snow

subsurface later in the year. The minimum $\delta^{18}O$ in the snow subsurface also occurs about 6 months later than the minimum $\delta^{18}O$ in the precipitation (November and May, respectively).

We explained this integration effect for the vertical difference between the snow surface and subsurface in Section 4.2.1. To make this more clear and include the explanation of the shift in the $\delta^{18}O$ monthly mean maximum, we have improved the subparagraph l.680 forward in the manuscript as follows:

"[...]. The higher $\delta^{18}O$ and lower d-excess in the summer precipitation compared to the winter precipitation (Fig. 6) explains why the mean summer snow surface is enriched in $\delta^{18}O$ and has a lower d-excess than the snow subsurface (Fig. 4). **This integration process can also explain, at first order, why the maximum and minimum monthly mean $\delta^{18}O$ in the precipitation, the snow surface and the snow subsurface do not occur at the same time (January and May for the precipitation-weighted, Fig. 6c; February and October for the snow surface; and March and November for the snow subsurface, Fig. 3b).**"

In section 3.2.2, the authors find that the surface snow is relatively enriched in $\delta^{18}O$ during warm season (from November to April) due to the sublimation of surface snow. Indeed, the calculated surface moisture fluxes also indicate sublimation occurs mainly in summer (Fig. 2). Theoretically, stronger sublimation should cause more enrichment of $\delta^{18}O$ in surface snow. As a result, I would like to know if there exists a correlation between the surface snow $\delta^{18}O$ and the water vapor flux during the summer. Additionally, the slope between $\delta D$ and $\delta^{18}O$ in surface snow should be lower than the $\delta D$-$\delta^{18}O$ slope of precipitation. I suggest the authors to add above relevant information to further validate the sublimation process because it is a key post-depositional process in inland of Antarctica.

We partly agree with the reviewer's comment. In section 3.2.2, we indeed find that the snow surface is relatively enriched in $\delta^{18}O$ from November to April. However, in Section 4.2.1, we discuss that the mean seasonal difference between the two snow layers is most likely explained by the integration effect of the samples (see comment above and l.677 forward in the manuscript), and that it is the inter-annual variability in the difference between the two snow layers that might, on the other hand, be explained by sublimation fluxes (l.666 forward). We therefore only hypothesise on the effects of sublimation on the snow surface.

Nevertheless, we performed a correlation analysis between the snow surface $\delta^{18}O$ and the daily mean water vapor flux during the summer months (November to January) and found that they correlate positively with a coefficient of 0.41 (Spearman's correlation coefficient, p-value < 0.001). We also find a negative correlation between the snow surface d-excess and the daily mean water vapor flux of -0.49 (Spearman's correlation coefficient, p-value < 0.001). We argue that these moderate correlations don't confirm nor rule out the effect of water vapor fluxes, because the snow surface samples cover different time periods than the days of sample collection (and therefore the daily mean water vapor flux). In addition, we expect the snow surface $\delta^{18}O$ to be correlated to the water vapor fluxes due to their individual correlation to the atmospheric temperature (precipitation $\delta^{18}O$ correlated to temperature that contributes to the signal in the snow and water vapor fluxes dependent on the humidity gradient between the surface and the atmosphere, thus also dependent on the atmospheric temperature).

Similarly, because the precipitation and snow surface samples don't cover the same time period, we argue that comparing their $\delta^{18}O$-$\delta D$ slopes will not necessarily provide additional information on the effect of sublimation. If we look at the precipitation and snow surface samples from December and January (summer months) in the $\delta^{18}O$ vs $\delta D$ domain, we find that the $\delta^{18}O$-$\delta D$ slope for the snow surface is higher than the slope for precipitation (see Figure A below). However, we see that most of the snow surface samples are below the precipitation samples (i.e. with a lower d-excess), which indicates sublimation in the summertime.

[Figure]

Figure A: Precipitation and snow surface $\delta^{18}$O versus $\delta$D during summertime (December and January). All precipitation samples with a negative d-excess were discarded to exclude any imprint of sublimation on the precipitation isotopic composition (see Section 4.2.2 of manuscript).

We have included additional information on this point in the discussion, improving the text l.666 forward as follows:

"We showed that at Dome C, sublimation occurs during the summertime (Fig. 2a) and that there is a net annual sublimation of snow (Fig. 2b), which means that the surface snow isotopic composition must have been affected by these fluxes. **We find that in the $\delta^{18}$O vs $\delta$D domain, most of the snow surface samples collected in December and January are below the precipitation samples (i.e. with a lower d-excess) collected in the same months (not shown), indicating an effect of sublimation on the snow surface isotopic composition in the summertime. In addition,** we observe some inter-annual variability in the vertical difference between the snow surface and subsurface isotopic compositions (Fig. 4). [...]"

In section 3.3.2, the authors compared precipitation isotopic composition ($\delta^{18}$O and d-excess) with the simulations of ECHAM6-wiso. Although the daily $\delta^{18}$O modelled by ECHAM6-wiso shows a good agreement with the observations, the ECHAM6 overestimates (underestimates) the $\delta^{18}$O (d-excess) likely due to the warm bias of the model. In addition, the daily d-excess is poorly simulated by the ECHAM6-wiso (Fig. 7b). The possible deficiency of the ECHAM6-wiso for the mismatch between the simulations and observations should be brief explained for guiding scientists to better utilize GCMs-wiso.

We acknowledge that we only briefly touch upon the origin of the observed biases between the observations and ECHAM6-wiso (l.760-766) but don't suggest any improvements for the representation of isotopes in the GCM. As also suggested by reviewer 2, we have now included in the manuscript some suggestions for the modelling community (l.763 forward):

"A thorough investigation of the biases in ECHAM6-wiso is beyond the scope of this study, but they might arise from different processes in the model, such as the environmental conditions at the moisture source region, the moisture transport and pathway, the supersaturation parametrization or the condensation height and temperature at Dome C. **The fourteen-year record of the precipitation isotopic composition (Dreossi et al., 2023 and this study) gives the opportunity to evaluate isotope-enabled GCMs and can be used to improve the tuning of the empirical parameterization of supersaturation in polar clouds (e.g. Risi et al., 2013).**"

The authors conclude that the mean effect of post-depositional processes enriches surface snow $\delta^{18}$O by 3.3‰ to 6.6‰ and lowers the snow surface d-excess by 3.5‰ to 7.6‰. As the authors analyzed, summertime sublimation should contribute significantly to the mean effect. After comparing Fig. 3b and Fig. 6c, I found the $\delta^{18}$O values of surface snow during wintertime are also higher than the values of precipitation. This suggests that the post-depositional processes rather than sublimation are also significant in winter. So, what kind of post-depositional processes lead to the enrichment of surface

snow $\delta^{18}$O in winter? Is it the diffusion of water vapor? The authors should discuss the post-depositional processes in wintertime.

On the seasonal scale, we are careful to not overinterpret the difference between precipitation and snow surface due to several processes that are difficult to disentangle, specifically sublimation/condensation, interstitial diffusion, wind erosion and mixing and snow surface integration (see comment 1 for the latter). We therefore mainly make a statement on the overall mean difference (5 years) between the precipitation and the snow surface and are cautious to attribute this difference to the effect of one process only.

However, the reviewer's comment is a very interesting point that we briefly discuss in Section 4.2.1 (l.683 forward), but we agree that we miss a discussion on the post-depositional processes in the wintertime. We have improved the text in Section 4.2.1 as follows:

l.687 forward

"Indeed, the isotopic composition of the snowpack is affected by diffusion of water molecules along isotopic and temperature gradients (Johnsen et al., 2000; Gkinis et al., 2014). Implementing this process in the model would possibly reduce the too large amplitude of the seasonal cycles in the synthetic snow layers to match the observations (Fig. 8b and d, 9b and d). **Diffusion might also partly explain the difference observed at the seasonal scale between the precipitation and the snow surface isotopic compositions (Fig. 3b and d, Fig. 6c and f), smoothing the incoming high-amplitude seasonal signal of precipitation and leading to a snow surface enriched in $\delta^{18}$O (and with lower d-excess) in the wintertime compared to precipitation.** In addition, some of the short-term variations [...]."

l.704 forward

"[...]. This process was proposed by Casado et al. (2018) to explain the slow decrease in the surface snow $\delta^{18}$O during the winter, a pattern observed in our study (Fig. 3a and b) and that cannot be explained by precipitation only (Fig. 8a and b). **A recent laboratory study also suggests that snow metamorphism during wind transport ("airborne snow metamorphism") has the potential to impact the snow isotopic composition in both $\delta^{18}$O and d-excess (Wahl et al., 2024). This could be an additional process occurring at Dome C explaining some of the differences between the precipitation and snow isotopic signals, including in the wintertime.**"

References

Risi, C., Landais, A., Winkler, R., and Vimeux, F.: Can we determine what controls the spatio-temporal distribution of d-excess and [17]O-excess in precipitation using the LMDZ general circulation model?, Clim. Past, 9, 2173–2193, https://doi.org/10.5194/cp-9-2173-2013, 2013.

Wahl, S., Walter, B., Aemisegger, F., Bianchi, L., and Lehning, M.: Identifying airborne snow metamorphism with stable water isotopes, EGUsphere [preprint], https://doi.org/10.5194/egusphere-2024-745, 2024.

---

## Author Comment (AC2)

Response to anonymous referee #2

We thank Reviewer 2 for their time and effort to provide detailed and constructive feedback on the manuscript, which has improved the quality of the study. We have addressed all comments below and propose to implement the changes in a revised version of the manuscript.

Black: reviewer comment
Blue: author's response
Green: revised text (**bold: new text**)

The authors present an extensive analysis of existing and new data of stable water isotopes from precipitation, surface and subsurface snow samples collected next to the Dome C research station in Antarctica over the period from 2017 to 2021. They complement this data with a modelling approach – the Snow Isotopic Signal Generator (SISG) – which tries to simulate the observed isotopic composition with different input parameters. Additionally, they analyse the output from an isotope-enabled GMC (ECHAM6-wiso). The authors put a lot of effort into combining different isotope data sets from different campaigns. This data is very valuable for the community. I appreciate the dedication of this team to collect and measure these samples and to provide an in-depth analysis to
the community. Such data and the knowledge that the community gains from it are a necessary step to better understand the post-depositional processes on ice sheets and the climatic imprint in ice cores.

Specific comments (major)

The manuscript reads lengthy in parts and explains very detailed the observations of precipitation, surface and subsurface snow isotopic composition as well as the modelling approach. In parts it is difficult to follow and to remember all details from the comparisons of different observations and the modelling results. The readability can be improved by shortening parts. Additionally, chapter 4.1 (*Contribution of precipitation to the snow isotopic composition*) shows even more new figures and information. To me, (part of) this section provides many results and does not necessarily discuss them. The authors might consider moving this (or part of this) part to the results section and focus on the comparison of their findings to other studies as well as on implications of their findings in the discussion (as they do from chapter 4.2 onwards).

Many thanks to the reviewer for these useful suggestions. We partly agree that Section 4.1 would fit better in the results section of the paper, but we see the argument of the reviewer that it would improve the organisation of the manuscript. We therefore propose to move the whole Section 4.1 to the results, as a new Section 3.4 keeping the title "Contribution of precipitation to the snow isotopic composition". The discussion section will now be focusing on the comparison of our findings to other studies as well as on implications of our findings, as suggested by the reviewer.

Additionally, I would like to see comments or a chapter about the limitations of the study. Some limitations are mentioned in some parts, but I am missing a comprehensive analysis of the shortcomings and the reliability of their dataset and, especially, the assumptions on which their study is based on (e.g. isotopes represent local air temperature, scaling of ERA5 precipitation, large surface features and intermittent accumulation at their study site, etc).

Following the previous comment, we propose to reorganise the discussion and add a subsection focusing on the shortcomings and reliability of our dataset. The discussion is now organised as follows (Section 4.2. and onwards remain the same):

**4.1. Limits of the methodological approach**
    **4.1.1. Reliability of the datasets**
    4.1.2. SISG setup and inputs

This new discussion will incorporate the reviewer's major comments addressed below. The new text in the subsections 4.1.1 and 4.1.2 is available below the next comment.

- Figure 8, for example, show a large variability between consecutive samples. This can be related to a large spatial heterogeneity in surface features (e.g. Picard et al., 2019), likely causing variability and noise in surface and subsurface snow isotopic compositions. How do you explain these differences? How realistic is it that these two samples, 50m apart, sampled twice a week, represent the local weather/climate, considering the reported surface processes and post-depositional processes (e.g. snow erosion and re-deposition)?

We agree with the reviewer's comment on the large variability between consecutive samples, and that this can be indeed due to large heterogeneity in surface features. We are discussing this point in Section 4.2.1 (l.693 forward) as a potential explanation for the short-term discrepancies between the snow surface signal and the incoming precipitation signal (represented as the synthetic snow). Here we consider erosion and re-deposition processes as one of the post-depositional processes that affects the snow isotopic composition.

We acknowledge the reviewer's concern that we don't discuss the reliability of using two samples collected 50 m apart and twice a week as a representation of the local weather and climate. In the manuscript we are only showing the average between the two sampling locations to describe the variability in $\delta^{18}O$ and d-excess in the snow surface and subsurface. In order to understand whether this is realistic, we looked at the temporal variations of the two samples independently (locations 1 & 2, see Figure A1 below). We see that the variations in $\delta^{18}O$ and d-excess at locations 1 and 2 generally follow each other, particularly for the snow surface samples and less clear for the snow subsurface samples. To go beyond the visual comparison and to determine the timescales at which the two samples show the same signal, we performed a wavelet coherence analysis (Grinsted et al., 2004) between the time series from the two locations (see Figure A2 below). We find a significant in-phase coherence between the two locations in the range of approximately 120 days and above throughout the 5-year period, for both $\delta^{18}O$ and d-excess in the surface and subsurface samples. Note that this is a bit less clear for the d-excess in the subsurface samples. We also episodically see some coherence between the two locations at 30 to 60 days.

This analysis shows that the two samples collected 50 m apart present the same variations in $\delta^{18}O$ and d-excess from intra-annual (sometimes down to 30 days) to multi-annual timescales, but not at shorter timescales. We argue that this shared signal between the two locations indicates that the variations in $\delta^{18}O$ and d-excess observed in the snow samples do represent the local weather and climate driven isotopic signal in surface snow at intra-annual to multi-annual timescales. At shorter time scales (intra-monthly), it is most probably sampling noise that dominates the variations in the snow $\delta^{18}O$ and d-excess, which can be induced by erosion and redistribution and reflects more the spatial variability than the temporal one. This short-term noise and variability are also the reason why we are not quantifying the post-depositional effects on intra-monthly to seasonal timescales, but solely the mean effect on the snow over five years.

We have included this analysis in the manuscript in a new Section 4.1.1 of the discussion (Reliability of the datasets, see text below).

[Figure]

Figure A1: Snow isotopic compositions of the two samples collected 50 m apart (locations 1 & 2). Panels (a) and (b) show the time series for the snow surface and subsurface $\delta^{18}$O, respectively. Panels (c) and (d) show the time series for the snow surface and subsurface d-excess, respectively. In all four panels, the thin lines represent the original time series and the thick lines show the 30-day running means.

[Figure]

Figure A2: Squared Wavelet Coherence analysis between the two isotopic composition time series of the samples collected 50 m apart (locations 1 & 2) using the procedure of Grinsted et al. (2004). Panels (a) and (b) show the analysis for the snow surface and subsurface $\delta^{18}$O, respectively. Panels (c) and (d) show the analysis for the snow surface and subsurface d-excess, respectively. In all four panels, the 5% significance level against red noise is shown as a thick contour. The light white shade shows the cone of influence where edge effects might distort the picture. Arrows indicate phasing (angle corresponds to phase behaviour). The vertical black lines represent the first day of each year in the 5-year time series. Note that we linearly interpolated the four time series to daily time steps to be able to perform the analysis.

To address the reviewer's previous comments, we have implemented changes in the discussion which are provided below. Both figures A1 and A2 above were added to the Supplementary Materials (new figures S2 and S3).

[revised manuscript text omitted]

- Sublimation is often discussed as a reason to alter the surface snow isotopic composition significantly throughout the 5-year period. The authors discuss the difference between surface and subsurface snow isotopic composition, but do not use or apply the mentioned model by Wahl et al. (2022) to simply estimate the depth of snow affected by sublimation. Moreover, they mention (l. 695ff.) that the location at Dome C on the East Antarctic Plateau has a very intermittent snow accumulation in time and space (which is well-known). Besides presenting the new datasets, this study does not provide more insights into the mechanism of post-depositional processes affecting the stored climatic information in stable water isotopes in such environments. Extending their simulations with e.g. diffusion and/or sublimation (models for both do already exist, (e.g. Münch et al., 2016; Wahl et al., 2022)) or simply estimating the effect of these processes, would help to quantify the role of these processes and to further disentangle ambient noise, post-depositional processes, and the climatic imprint in stable water isotopes in precipitation as well as in surface and subsurface snow.

We acknowledge that our study primarily focuses on the observational datasets from Dome C without applying any complex model to study the impact of the different post-depositional processes on the snow isotopic signals. However, we believe that the datasets presented are quite extensive and significant and therefore required a thorough investigation and discussion about their reliability, as also suggested by the reviewer. In addition, although we do not quantify its impact on the snow isotopic composition, we are providing a 3-year time series of water vapor fluxes at Dome C overlapping with the precipitation and snow samples collected on site, which paves the way for future studies investigating the role of water vapor fluxes on the snow isotopic composition at Dome C. The use of a more complex model and a full observation-model comparison is beyond the scope of this manuscript, but we agree with the reviewer that this is a necessary step forward and are working on a follow up study to further disentangle the processes building up the isotopic signal in the snow.

- ECHAM6-wiso is compared to the observations and the SISG simulations. What is the benefit of including a GCM here? Can you provide any suggestions for the modelling community how to improve the representation of isotopes in isotope-enabled GCMs? Why not using a smaller model, such as Wahl et al. (2022) or Dietrich et al. (2023) developed when they proved that they can reproduce observations?

In our study, the outputs from ECHAM6-wiso (precipitation amounts and isotopic composition) are first evaluated against observations and later on used as inputs to the simple SISG model. We included a GCM in our study for two reasons: (1) investigate possible biases and evaluate the model performance in representing the observed precipitation amounts (Section 3.3.1) and isotopic compositions (3.3.2) and (2) use the model outputs first in a simple model to compare the results with alternative inputs (Section 4.1.1 and 4.1.2).
We would like to stress that the modelling approaches in Wahl et al. (2022) and Dietrich et al. (2023) are used in two different contexts with two different versions of the SNOWISO model (latest version in Dietrich et al., 2023; see also our response to the reviewer's last comment below) and that in Dietrich et al. (2023), the model is driven by (but not only) outputs from a GCM. In our study, the aim was therefore to prepare the foundation for using GCM outputs as drivers of a snowpack model to simulate the effects of post-depositional processes on the isotopic signal found in the snow and firn, as done by Dietrich et al. (2023) for the Greenland Ice Sheet.
We acknowledge that we only briefly touch upon the origin of the observed biases between the observations and ECHAM6-wiso (l.760-766) but don't suggest any improvements for the representation of isotopes in the GCM. As also suggested by reviewer 1, we have now included in the manuscript some suggestions for the modelling community (l.763 forward):

"A thorough investigation of the biases in ECHAM6-wiso is beyond the scope of this study, but they might arise from different processes in the model, such as the environmental conditions at the moisture source region, the moisture transport and pathway, the supersaturation parametrization or the condensation height and temperature at Dome C. **The fourteen-year record of the precipitation isotopic composition (Dreossi et al., 2023 and this study) gives the opportunity to evaluate isotope-enabled GCMs and can be used to improve the tuning of the empirical parameterization of supersaturation in polar clouds (e.g. Risi et al., 2013).**"

Specific comments (minor)

- L. 59: lower atmosphere

Taken into account.

- L. 81: water stable isotopes -> stable water isotopes

Taken into account.

- L. 81f.: It is written "snow surface and subsurface" and later "surface snow and subsurface snow". Please consistently use one term throughout the manuscript.

To be consistent we now only use the term "snow surface and subsurface" throughout the manuscript.

- L. 118: Is the uncertainty determined with an independent quality control sample?

The uncertainty stated here is estimated using sample replicates, standards with known isotopic composition and independent control samples. Please see two comments below for the text modification.

- L. 118: two standard deviations

This will now become "one standard deviation" (see comment below).

- L. 118 and l. 142: First, the uncertainty is given by two standard deviations and later, in l. 142, by one standard deviation of a quality standard. Why are you using different measure for uncertainty? It seems that the first uncertainty is lower when using one standard deviation, especially for dD (0.7 ‰/ 2 = 0.35 ‰). Does this influence the d-excess presented in this study?

The uncertainties associated to the measurements of the snow and precipitation samples were estimated differently:
- Snow samples: the uncertainty is estimated using sample replicates, standards with known isotopic composition and independent control samples.
- Precipitation samples: the uncertainty is estimated by measuring several times a quality standard with known isotopic composition.

We agree with the reviewer that the text was misleading and there was also a mistake in the original manuscript. The uncertainties stated for the snow samples correspond to one standard deviation. We will keep one standard deviation for both the snow and precipitation samples, and round the uncertainty values of the precipitation samples (see modified text below).

From the uncertainties on $\delta^{18}$O and $\delta$D, we can estimate the uncertainties (one standard deviation) on d-excess, which is ± 0.9‰ for the snow samples and ± 0.8‰ for the precipitation samples. This doesn't influence the d-excess presented here, with variations of d-excess in the precipitation and in the snow higher than the uncertainties stated above.

To incorporate the reviewer's comment, we have improved the text as follows:

l.118-119
"The associated uncertainty (one standard deviation including quality control sample, standards and sample replicates) on these measurements is ± 0.2‰ for $\delta^{18}$O and ± 0.7‰ for $\delta$D, **which yields an uncertainty of ± 0.9‰ for d-excess (one standard deviation).**"

l. 141-143
"The associated uncertainty (one standard deviation of quality standard replicates) on these measurements is ± 0.1‰ for $\delta^{18}$O and ± 0.7‰ for $\delta$D, **which yields an uncertainty of ± 0.8‰ for d-excess (one standard deviation).**"

- L. 151: The time step is one hour. How is a 1-hour average calculated from a 1-hourly spaced timeseries?

We agree that this is confusing. Although the weather station measures more frequently, only the 1-hourly averaged data is available in open-access. We have modified the text l.151 as follows:
"Hourly data from the AWSIT is available in Grigioni et al. (2022)."

- L. 160: Why are you not averaging the data to 1-hour time steps as well (as the AWSIT data)?

We wanted to have the highest time resolution for the water vapor fluxes, and since the data from the USt is available in open access at 30 min time step, we chose this resolution instead of 1-hour time steps.

- L. 167 and 169: Which test was used to test for significance?

It is a misuse of the term "significant" because we didn't perform any statistical test. We agree that this is an issue and therefore performed a correlation on the overlapping period and found a correlation (Pearson correlation coefficient) of 1.0. We have modified the text l.167 as follows:
"We found a linear correlation (Pearson correlation coefficient) of 1.0 between the datasets for both temperature and wind speed. Therefore, we use the temperature and wind records from the USt between 2017 and 2021 in our analysis."

We notice that similarly, in l. 169 we use the term "no significant difference" without having performed a statistical test. We have modified this sentence as well as follows:
"The linear correlation (Pearson correlation coefficient) between the two temperature records is 0.99 (not shown)."

- L. 191: flux

Taken into account.

- Table 1: The sampling rate of the CALVA measurements is given as 30s. Here, you are referring to the used averages. "Measurement time step" is in my opinion not the correct term here. Maybe you can rephrase it.

This is a good point. We have changed the term "measurement time step" in the table caption by "averaging time step".

- L. 287: Which relationship is used here to calculated d18O and dD from atmospheric temperature?

We use the linear relationships between $\delta^{18}O$ and $\delta D$ of the precipitation samples and the daily mean temperature at 3 m, as stated in the sentence l.286-287: "The precipitation isotopic composition is calculated from the atmospheric temperature, using the linear relationships between $\delta^{18}O$ and $\delta D$ of the precipitation samples and the 3 m daily mean temperature (Eq. 1 and 2 in Sect. 3.3.3)."

- L. 460: How do you explain the fact that the precip-weighted means are higher than then arithmetic means?

The precipitation-weighted overall $\delta^{18}O$ means are higher than the overall arithmetic means, for both the observations and the model, because the lowest $\delta^{18}O$ values in precipitation are associated with smaller precipitation amounts (skewed distribution, Figure A3 below), and therefore weigh less when computing the precipitation-weighted averages.

[Figure]

Figure A3: Daily precipitation δ¹⁸O versus daily precipitation amounts, for the observations in panel (a) and the ECHAM6-wiso outputs in panel (b). In both panels, the solid black line and dashed grey lines show the 25$^{th}$, 50$^{th}$ and 75$^{th}$ percentiles.

We have included this explanation into the discussion (Section 4.2.2) improving the text l.724 forward as follows:

**"As shown in Section 3.3.2, the observed δ¹⁸O precipitation-weighted mean over five years is higher than the overall mean (-53.4 and -56.2‰, respectively). This is explained by the lowest δ¹⁸O values in precipitation being associated with smaller precipitation amounts (Fig. S8 in Supplement S6), and therefore weigh less when computing the precipitation-weighted average. The opposite applies for d-excess: the weighted overall mean is lower than the arithmetic overall mean (12.2 and 15.2‰, respectively) because high d-excess values are associated with lower precipitation amounts (Fig. S8 in Supplement S6).**

**Now because the snow surface layer represents the amount of snow accumulated over a certain period, its mean isotopic composition should reflect the weighted mean (by precipitation amounts) isotopic composition of precipitation. Therefore,** we express the mean effect of post-depositional processes at Dome C as the difference between the observed 5-year weighted mean isotopic composition of precipitation and the observed 5-year mean isotopic composition of the surface snow.

Considering all precipitation and snow samples, the surface snow δ¹⁸O is 2.4‰ higher than in precipitation and the surface snow d-excess is 1.8‰ lower than in precipitation (Table 6). [...]"

- L. 469f.: You write that "the summertime" difference between observations and ECHAM6-wiso "decreases when the precipitation d18O is weighted by the precipitation amounts". To me, this only accounts for December and January while the overall difference increases. Can you explain why? What is your conclusion here?

Both the observations and the model have higher precipitation amounts associated with higher δ¹⁸O (see previous comment), but the δ¹⁸O modelled by ECHAM6-wiso is in general higher than the observations (Fig. 6b) and high modelled δ¹⁸O values are associated with larger precipitation amounts than for the observations (Figure S8a and c below). Therefore, when weighted by precipitation amounts, the δ¹⁸O modelled by ECHAM6-wiso is increased more than the observations, leading to an increase in the overall difference.

For the summer months December and January, high and low δ¹⁸O values modelled by ECHAM6-wiso are associated with similar precipitation amounts (see Figure S8a and c below), while for observations, higher δ¹⁸O values are associated with higher precipitation amounts (similarly as for rest of the year). Therefore, in summer, the high and low δ¹⁸O values from ECHAM6-wiso weigh the same when doing the monthly weighted average, reducing the discrepancy with observations. This means that there is a compensation effect between the modelled δ¹⁸O and precipitation amounts during summertime. The opposite applies to d-excess: when computing the weighted monthly means, the overall difference decreases while the summertime difference increases (Fig. 6f, see also previous comment). This is explained by the distribution of d-excess values against precipitation amounts (Figure S8b and d below) in addition to modelled d-excess values lower than observations (Fig. 6e).

We have included this information into the Supplementary Materials and added a brief explanation in the discussion l.760 forward as follows:

"The weighted-mean isotopic composition of precipitation modelled by ECHAM6-wiso over the whole period is 7.7‰ higher in $\delta^{18}O$ and 7.1‰ lower in d-excess than the observed precipitation (Table 6, Fig. 6c and f). These large differences result from the biases identified in the model (Sect. 3.3.2, Fig. 6 and 7), **combined with the distribution of the daily isotopic values against precipitation amounts (Fig. S8 in Supplement S6), which influences both the overall mean and the seasonal difference between the observations and the model (Fig. 6 and Supplement S6).** This shows the limitations of using ECHAM6-wiso simulations [...]"

To accompany the new text implemented as our answer to the two previous comments, we also added the following figure in the Supplementary materials of the paper:

[Figure]

**Figure S8: Daily precipitation isotopic composition ($\delta^{18}O$ in panels (a) and (c), d-excess in panels (b) and (d)) versus daily precipitation amounts, for the observations in panels (a) and (b) and the ECHAM6-wiso outputs in panels (c) and (d). In all four panels, the solid black line shows the mean isotopic composition of daily precipitation and the coloured markers (brown in panels (a) and (c) and red in panels (b) and (d)) show the days in December and January.**

- Figure 7:
  - Why are you only reporting the RMSE and not the correlation as well?

This is an oversight and we agree that the correlation coefficient is a valuable additional metric to evaluate the performance of the model. The Pearson correlation coefficients are now included in Figure 7, and the text below the figure has been modified accordingly:

l. 508-509
"The daily precipitation $\delta^{18}O$ modelled by ECHAM6-wiso shows a good agreement with the observations, with a linear regression slope of $0.84 \pm 0.03$ (1.0 being the perfect fit), **a Pearson correlation coefficient of 0.65** and a root mean square error (RMSE) of 8.8‰ (Fig. 7a)."

l.510-512

"For d-excess, the ECHAM6-wiso model results only poorly represent the observations, with a linear slope of 0.1 ± 0.02, **a Pearson correlation coefficient of 0.17** and a RMSE of 16.4‰ (Fig. 7b)."

- I assume that the data shown in the scatter plot are the same as in Figs. 6a and 6d. It seems that ECHAM6-wiso suggests more days with precipitation than the observations do. How are you handling this? Are you accounting for these or are you ignoring these days?

It is correct that the scatter plots in Figure 7 show the data from Figure 6a and b. However, there are gaps in the observed time series of the precipitation isotopic composition. This can be due to several reasons: either the sample collected on site was too small to perform an isotopic analysis on it, either the precipitation fallen on the sampling table was blown away and no sample could be collected, or the sample was simply not collected in the field (due to harsh weather for example). On the other hand, ECHAM6-wiso models the isotopic composition of precipitation for everyday over the period. As explained in Section 2.5.2 (l.296f), the model outputs are post-processed, so some days in the time series are also associated with missing isotopic values. We can only compare the days where we have both the observed and modelled precipitation isotopic composition, so the days without observation and/or model output are not included here. This means that in Figure 7a, 886/1826 days are shown and in Figure 7b, 882/1826 days are shown. We have updated the figure caption as follows:

"Figure 7. Daily observed precipitation $\delta^{18}$O (a) and d-excess (b) versus daily modelled precipitation isotopic composition at Dome C. All precipitation samples collected between 2017 and 2021 are shown (described in Sect. 2.3), together with the corresponding daily values from ECHAM6-wiso simulation results (described in Sect. 2.5). **Note that due to missing observations and post-processing of ECHAM6-wiso outputs, 886/1826 days are shown in panel (a) and 882/1826 days in panel (b)**. The coloured lines correspond to the linear fits between the observations and the model results (slope coefficients given in legends, both linear regression coefficients are significant with a p-value < 0.001)."

On the post-processing of the ECHAM6-wiso outputs, we have omitted to mention an important point: all daily precipitation rates below 0.0016 mm w.e. per day (= 0.05 mm w.e. month$^{-1}$) are set to zero, and the associated isotopic composition of precipitation is set to "missing value". This is to prevent unrealistic values in the precipitation isotopic compositions due to very low precipitation amounts (numerical effect). As East Antarctica is a very dry region, the threshold was chosen to be ten times lower than the one commonly used for rain gauges (0.5 mm w.e. month$^{-1}$, used for example on the dataset from the Global Network of Isotopes in Precipitation). We have updated the text when describing the ECHAM6-wiso simulation as follows:

l.269 forward

"The fifth experiment uses the precipitation amounts given by the isotope-enabled global circulation model (GCM) ECHAM6-wiso (described in Cauquoin et al., 2019). The simulation was performed at a spatial resolution of 0.9° and nudged to ERA5 reanalyses (Cauquoin and Werner, 2021). The daily precipitation amounts were extracted for the grid point closest to Dome C. **As the days with very low precipitation rates are not considered as "precipitation days", any precipitation rate below 0.0016 mm w.e. day$^{-1}$ is set to zero. Because East Antarctica is a very dry region, this threshold was chosen to be ten times lower than the one commonly used for rain gauges (0.5 mm w.e. month$^{-1}$, e.g. used for the Global Network of Isotopes in Precipitation data). Note that such a threshold was not applied to the precipitation amounts given by ERA5."**

l.296 forward

"The fifth experiment uses the daily precipitation isotopic composition modelled by ECHAM6-wiso. **To prevent any unrealistic values because of a numerical effect when the precipitation amounts are very low, the days with precipitation amounts below 0.0016 mm w.e. (Section 2.5.1) were associated with missing values of the precipitation isotopic composition. In addition,** all data points outside of the 5-year average ± three standard deviation range were discarded (six data points for $\delta^{18}$O and 20 data points for d-excess over five years). A comparison of the observed daily precipitation isotopic composition and ECHAM6-wiso simulations is shown in Fig. 6 (Sect. 3.3.2)."

- L. 528f: Are these the equations used in the ECHAM6-wiso model itself or are these empirically calculated?

Equations (3) and (4) are not used/coded in the model itself but calculated from the ECHAM6-wiso outputs of the precipitation δ¹⁸O and δD and the atmospheric temperature to compare with the relationships we find in the observations.

- L. 811ff.: My understanding of a proxy system model might be different than yours. Following Evans et al. (2013), for instance, a proxy system model may be defined as the complete set of forward and mechanistic processes by which the response of a sensor to environmental forcing is recorded and subsequently observed in a material archive. Wahl et al. (2022) and Dietrich et al. (2023) did a great job in coming up with statistical approaches to e.g. calculate expected isotopic variations in surface and subsurface snow from sublimation. However, I see their models rather as an input to more sophisticated (proxy system) models.

We acknowledge the reviewer's comment that the model initially developed by Wahl et al. (2022) for the surface snow and further developed by Dietrich et al. (2023) to simulate firn cores does not entirely meet the definition of a proxy-system model by Evans et al. (2013). Nonetheless, the latest version of the model (SNOWISO model, Dietrich et al., 2023) can be considered as a first step toward such a proxy-system model. We have slightly modified the text l.811 forward as follows:

"To progress towards an accurate quantitative interpretation of isotopes in ice cores, we recommend developing a *proxy system model* **(Evans et al., 2013)** that can be coupled to isotope-enabled GCMs. **A first step toward** such a model has been developed recently by Wahl et al. (2022) and Dietrich et al. (2023) for Greenland and includes mechanical processes leading to the recording of isotopes at the ice sheet's surface and in firn cores. The datasets presented in our study can therefore be useful to calibrate and validate this kind of model at intra-annual, seasonal, and inter-annual timescales at the deep drilling site of the EPICA Dome C ice core."